# The role of repetitive thinking and spirituality in the development of posttraumatic growth and symptoms of posttraumatic stress disorder

Catrin Eames[¤a]*, Donna O'Connor[¤b]

Institute of Psychology, Health and Society, University of Liverpool, Liverpool, United Kingdom

¤a  Current address: School of Psychology, Faculty of Health, Liverpool John Moores University, Liverpool, United Kingdom
¤b  Current address: Greater Manchester Mental Health NHS Trust, Department of Clinical Psychology, North Manchester General Hospital, Manchester, United Kingdom
*  c.e.eames@ljmu.ac.uk

**Data Availability Statement:** Data cannot be shared publicly because of limitations of consent when collecting data. Data are available from the University of Liverpool Institutional Data Access or

## Abstract

Both post-traumatic growth (PTG) and post-traumatic stress disorder (PTSD) are associated with spirituality and different kinds of repetitive thinking, such as deliberate rumination (DR) and intrusive rumination (IR), respectively. This study aimed to examine if spirituality modifies the relationship between types of rumination and trauma outcomes. Ninety-six students completed an online survey of four questionnaires in a cross-sectional online survey: The Posttraumatic Stress Diagnostic Scale, the Event Related Rumination Inventory, the Posttraumatic Growth Inventory-Short form and the Expressions of Spirituality Inventory-Revised. Findings revealed that spirituality was related to DR and PTG, but not to IR or PTSD symptoms. Moderation analysis showed that spirituality significantly moderated the relationship between PTG and DR, but not the relationship between PTSD and IR. These findings indicate that while spirituality has no relationship with negative outcomes of trauma, it may help individuals to ruminate in a constructive manner in order to develop positive outcomes.

## Introduction

Positive and negative psychological characteristics and aspects of existence exist on a continuum, often interact clinically, and positive characteristics can be considered buffers against the development of distress [1]. One area of psychological research in which positive factors are emerging is that of trauma, where, in addition to the negative consequences experienced by individuals following a trauma, there is also the potential for positive psychological change to occur.

### Trauma

A traumatic event may be defined as one in which an individual directly or indirectly experiences or is witness to an event(s) that involves actual or threatened death or serious injury to

via contacting recman@liverpool.ac.uk or
ethics@liverpool.ac.uk

**Funding:** The authors received no specific funding
for this work.

**Competing interests:** The authors have declared
that no competing interests exist.

self or others and must elicit a response of intense fear, helplessness or terror [2]. Trauma can
be categorised as Type I, i.e. sudden, short-term events such as earthquakes, and Type II
trauma, i.e. incidents that are chronic and sustained, such as childhood sexual abuse. Regard-
less of event type, the experience of a trauma can invalidate an individual's existing under-
standing and view of the world and can result in extensive impact on a person's identity [3].
The subjective severity of the event activates emotional distress. Often the distress this causes
can prompt individuals to try to make sense of the meaning of the traumatic event(s) [4, 5].
Failure to find meaning has been found to be associated with poor psychological outcomes,
including depression and post-traumatic stress disorder (PTSD) [6]. PTSD may be defined by
three clusters of symptoms: the re-experiencing of the traumatic event, persistent avoidance of
trauma-related stimuli, and heightened persistent symptoms arousal. Memory, affect and
response to trauma-stimuli are all effected, with the re-experiencing of the event (e.g.,
experiencing flashbacks) often with intense emotional experience (e.g., panic) considered a
central component of PTSD [2].

## Post-traumatic growth

In contrast, when people find meaning in the aftermath of trauma, they have the potential to
experience profound positive psychological transformation and higher levels of psychological
functioning than prior to the event(s). This process is known as Post-Traumatic Growth
(PTG). Tedeschi and Calhoun [7] identified five domains of life in which the positive effects of
PTG may occur: improvements in relationships, a greater appreciation for life, new opportuni-
ties in life, a greater sense of personal strength, and spiritual development. PTG and PTSD are
not mutually exclusive and often co-exist, as an individual can experience distress whilst simul-
taneously growing [8]. Some report a linear relationship between the two constructs, where
the level of psychological growth experienced is directly proportional to the severity of the
trauma [9]; whilst others indicate that a curvilinear relationship may exist in which an opti-
mum level of suffering is required for growth, but excessive levels of distress may impede its
development [10, 11].

   PTG has been associated more with incidents of Type I trauma, rather than Type II [12].
PTG has been shown to occurr following a wide range of Type I traumatic events, including
natural disasters [13] road traffic accidents [14] illnesses [15], and interpersonal violence [16].
This may be due to the theoretical assumption that events need to be of significance to trigger
cognitive rumination, processing, and emotion regulation [7], but not so significant (for exam-
ple, repeated and prolonged trauma) as to hinder the process [17]. Culture has been found to
influence levels of PTG, with participants from the USA reporting higher levels of PTG than
other countries, possibly due to social desirability of responding to challenges with positivity
[18]. For example, the social pressure to report growth from adversity is greater in trauma sur-
vivors from the USA than those in Europe, and may therefore be a cultural expectation [18].
This is noted regardless of social factors such as non-Caucasian ethnicity and lack of educa-
tion, both of which are associated with higher growth and are risk factors for PTSD, whereas
lack of social support is associated with less growth [16]. Minority ethnic groups report higher
levels of PTG than white populations; partially attributed to being mediated by religiosity (e.g.
African-American and Hispanic populations; [19]). Some report an association between PTG
and age, but differ in terms of the direction of the relationship [20, 21] while others report no
significant relationship [22]. Women are more likely to experience PTG than men [23], and
student populations have been found to experience similar levels of traumatic events as the
general public [24]. The relationship between PTG and time since trauma (TST) appears
poorly understood, with some reporting a positive relationship (e.g. [25]) and others finding

none [26]. Those who report an association state conflicting time frames of emergence; however most agree that it takes time, sometimes many years, for PTG to develop [27].

## Cognitive processes and post-traumatic growth

For PTG to occur, the event must be 'seismic' enough to threaten previously held core beliefs, and it is the distress caused by the resulting cognitive dissonance and the "struggle with the new reality in the aftermath of trauma that is crucial in determining the extent to which post-traumatic growth occurs" [6, p. 5]. Individuals may initially find themselves experiencing 'intrusive rumination' (IR), described as being unwanted, repetitive thoughts about the traumatic event, which are often resisted, involuntary, difficult to control, and associated with attempts to avoid the thoughts [28]. IR is a necessary diagnostic criterion for PTSD [29], to include intrusions of persistently re-experiencing via memories, nightmares, flashbacks, emotional distress or physical reactivity after exposure to traumatic reminders [2]. These intrusive thoughts focus on negative aspects and affect of the event and can then prompt the individual to process the trauma in a more conscious and voluntary cognitive manner via somatic cues; this type of thinking has been termed 'deliberate rumination' (DR, [28]). It is a form of repetitive, active, reflective thinking about the event and its meaning that promotes the capacity to reflect on and re-assess one's existing schema and assumptions in light of the new experience and rebuild one's view of the world [30]. DR is considered a crucial cognitive mechanism for both meaning making and the development of PTG [31]; conversely, low levels of engagement in DR is associated with greater post-traumatic distress [32]. In line with the literature related to PTG, we adopt the terminology of deliberate and intrusive rumination throughout the manuscript. Here, DR is defined as a cognitive process that involves deliberately re-examining an event, that may promote finding meaning in an experience, helping to develop understanding, value and significance from an event. We acknowledge that deliberate reflection, the process of purposefully thinking about an experience with an intent to learn, can also be applied within this context.

## Spirituality

Belief systems can provide frameworks that accommodate challenging life circumstances into one's existing core beliefs [33]. Studies report mixed results, with belief systems associated with both positive and negative health outcomes (e.g. [34]). However many of these studies examined religious beliefs and Bryan, Graham and Roberge [35] argue the need for differentiation between religion and spirituality. Although spirituality is a difficult concept to define, Pargament [36] recommends that for the purpose of research, a working definition of "the search for the sacred" (p.12) be used. MacDonald et al. [37] expand on this definition: "spirituality is a natural aspect of human functioning which relates to a special class of non-ordinary experiences and the beliefs, attitudes, and behaviours that cause, co-occur, and/or result from such experiences. The experiences themselves are characterized as involving states and modes of consciousness which alter the functions and expressions of self and personality and impact the way in which we perceive and understand ourselves, others, and reality as a whole" (p.5). Spirituality has been found to be a moderator of distress, with higher levels of spirituality associated with lower levels of distress, and lower levels of spirituality associated with higher levels of distress [38], as spirituality can be seen as a source of support and guidance in the face of adversity.

## Current study and hypotheses

The study explores mechanisms that can promote adaptive outcomes following exposure to traumatic events to inform clinical practice. The interactive relationships of spirituality with

the subtypes of repetitive thinking in relation to PTG and PTSD are also explored. The current study evaluates whether 1) high levels of DR and high levels of spirituality are associated with PTG, 2) spirituality modifies this relationship; and 3) whether the relationship between IR and PTSD is moderated by spirituality. For those who hold spiritual beliefs as important, spirituality may provide a framework from which growth may occur via an interplay of cognitive, emotional, and social processes to explore core beliefs and challenging life circumstances.

**Hypotheses relevant to PTG:.**

1. Controlling for demographic variables, Post Traumatic Stress Disorder (PTSD) and Intrusive Rumination (IR), it is predicted that Time Since Trauma (TST), Deliberate Rumination (DR) and spirituality will all show unique positive associations with Post Traumatic Growth (PTG).

2. It is predicted that spirituality will moderate the relationship between DR and PTG, with high levels of spirituality showing a stronger positive relationship compared to low levels of spirituality.

**Hypotheses relevant to PTSD:.**

3. Controlling for demographic variables, PTG, DR and TST, it is predicted that IR will show positive associations with PTSD. It is also predicted that spirituality will show negative associations with PTSD.

4. It is predicted that spirituality will moderate that relationship between IR and PTSD, with low spirituality showing a stronger positive relationship compared to high spirituality.

## Materials and methods

### Participants

Ninety-six eligible participants took part in an online survey and were included in the final data analysis (see Fig 1 for participant retention at each stage). The majority identified as female, White British, post-graduate, home University students (see Table 1). Participants ranged in age from 19–59 years ($M$ = 26.10 years, $SD$ = 7.71). The year in which the traumatic event occurred ranged from 1997 to 2016 ($M$ TST = 57.45 months [4 years and 9 months], $SD$ = 46.64) with the majority of traumas experienced after 2010 (67.4%).

**Inclusion criteria.** University students were eligible to take part in the survey if they were over 18 years and had experienced a traumatic life event after the age of 16 years. This age limit was chosen in an attempt to capture incidents of Type I trauma (rather than complex Type II trauma which is less associated with PTG), and to ensure that participants had cognitively processed the trauma as adults and not as children. The identified event experienced by the participant must meet Criterion A of the DSM-IV [2] definition of a traumatic experience.

**Exclusion criteria.** Participants who experienced a traumatic event within the past 4 months were excluded to avoid distressing respondents in the acute phase of trauma.

### Design, ethical approval and statistical power

The study was cross-sectional collected via an online survey. Ethical approval was granted (Ref: IPHS-1516-LB-145). No ethical issues related to taking part in the study were reported. An *a priori* power calculation was conducted using G-Power software. Three predictors (DR, IR & spirituality) plus four potential control variables (gender, age, ethnicity and TST) were

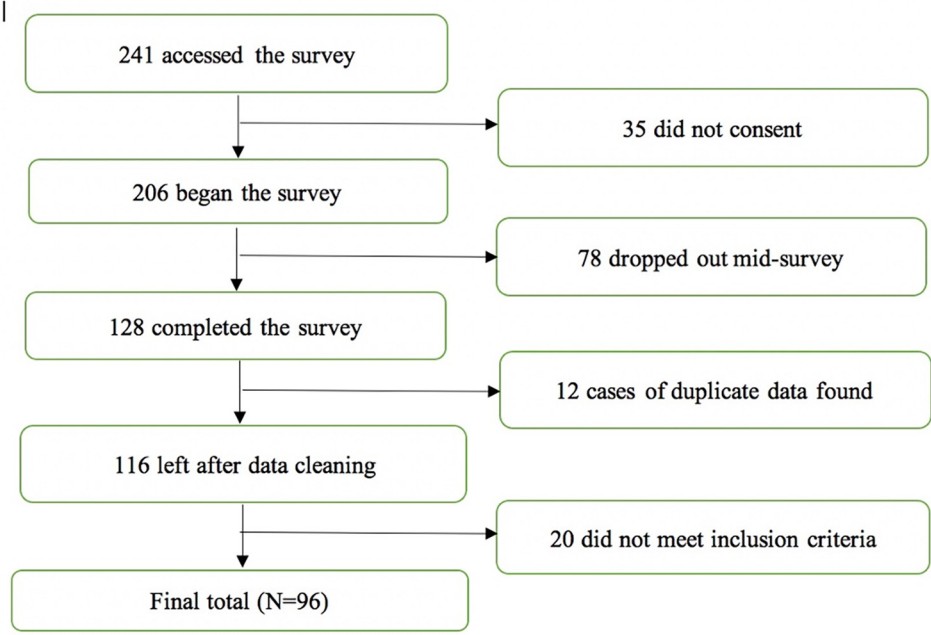

**Fig 1. Flow diagram of participation in the study.**

entered into the calculation. Based on a moderate effect size of $f^2 = 0.15$ [39], power of 0.8 and probability error of $\alpha = 0.05$, 103 participants would be required to achieve statistical power.

## Measures

**Demographic questionnaire.** Gender, age, ethnicity, student category (home or international) and year of study was asked of all participants.

**The posttraumatic stress diagnostic scale.** (PDS; [40]) gathers information about the nature of the traumatic event experienced. Part 1 checks that criterion A of the DSM-IV [2] for trauma is met by providing a checklist of 13 potentially traumatic events (PTEs), one of which is labelled 'other' with free-text option. Part 2 asks participants who select multiple traumas to indicate the most significant, with space to describe the event. It asks respondents how long ago the event occurred—in the present study this was changed to record specific month and year to increase accuracy. It also asks, with a 'Yes' or 'No' response, about resultant injury, belief about life-threatening situation, and whether they felt terrified. Two additional questions were included to ascertain whether they experienced a head injury during the event, and if they received any therapy in relation to the trauma. These were included as both factors were deemed important as potential confounders when measuring cognitive response to trauma. Part 3 of the PDS assesses levels of PTSD symptoms in response to exposure to the identified event where individual must experience at least one out of five listed symptoms of re-experiencing (Criteria B), at least three out of seven symptoms of avoidance (Criteria C) at least one out of five arousal symptoms (Criteria D). All three subscales are summed to assess symptom severity. The measure has been normed on a variety of trauma populations, has high face validity, good reliability ($\alpha = .92$ [37]) and good concurrent validity with the Impact of Event Scale ($r = .78$; [41]). Due to experimental error, item 39 ("How long have you experienced the problems that you reported above?") and part 4 ("Indicate if the problems have interfered with any of the following areas of your life") of the PDS were missing from the final

**Table 1. Description of participant demographics and trauma data.**

| | | *N* | % of sample |
|---|---|---|---|
| **Gender** | Male | 30 | 31.3% |
| | Female | 66 | 68.8% |
| **Ethnicity** | White British | 62 | 64.6% |
| | White Irish | 3 | 3.1% |
| | White Other | 9 | 9.4% |
| | British Asian | 5 | 5.2% |
| | Indian | 2 | 2.1% |
| | Pakistani | 2 | 2.1% |
| | Bangladeshi | 1 | 1.0% |
| | Chinese | 3 | 3.1% |
| | Asian (other) | 2 | 2.1% |
| | Black Caribbean | 1 | 1.0% |
| | Black British | 1 | 1.0% |
| | Black (other) | 1 | 1.0% |
| | White & Black Caribbean | 1 | 1.0% |
| | Other mixed background | 1 | 1.0% |
| | Arab | 2 | 2.1% |
| **Student category** | Home | 69 | 71.9% |
| | European Union | 7 | 7.3% |
| | International | 14 | 14.6% |
| **Year of study** | Undergraduate | 41 | 45.8% |
| | Masters | 23 | 24.0% |
| | Doctorate | 29 | 30.2% |
| **Potential traumatic event** | Serious accident | 37 | 38.5% |
| | Natural disaster | 11 | 11.5% |
| | Assault by someone known | 10 | 10.4% |
| | Assault by stranger | 21 | 21.9% |
| | Sexual assault by someone known | 7 | 7.3% |
| | Sexual assault by stranger | 8 | 8.3% |
| | Military combat | 3 | 3.1% |
| | Sexual contact while under-age | 5 | 5.2% |
| | Illness | 25 | 26.0% |
| | Other | 16 | 16.7% |
| **Met symptom criteria for PTSD** | Yes | 59 | 61.5% |
| | No | 37 | 38.5% |
| **Symptom severity of total sample** | Mild | 35 | 36.5% |
| | Moderate | 27 | 28.1% |
| | Moderate-severe | 26 | 27.1% |
| | Severe | 8 | 8.3% |
| **Symptom onset** | Immediately post event | 44 | 49.4% |
| | 1-week post event | 14 | 15.7% |
| | 1-month post event | 11 | 12.4% |
| | After 6 months | 8 | 4.5% |
| **Therapy received** | Yes | 19 | 19.8% |
| | No | 76 | 79.2% |

online survey and therefore were not included. In the current study, the PDS was found to have very good internal consistency (α = 0.93).

**The event related rumination inventory.** (ERRI; [28]) is a 20-item scale used to assess levels of rumination during the weeks immediately following a highly stressful event. Participants are asked to rate how often, if at all, they experienced thoughts related to the event (0 = not at all– 3 = often). The first 10 items measure intrusive rumination (IR) and the final 10 measure deliberate rumination (DR). The ERRI has been validated on a student sample and good internal reliability has been found for both intrusive ($\alpha$ = .94) and deliberate ($\alpha$ = .88) items, with both scales validated in relation to comparable variables, as well as correlating with posttraumatic distress and growth respectively. In the current study, the ERRI was found to have very good internal consistency ($\alpha$ = 0.93).

**The post traumatic growth inventory–short form.** (PTGI-SF; [42]) is a shorter, revised form of the original 42-item PTGI [43]. The PTGI-SF contains 10 items, two from each of the five PTG subscales. It asks participants to rate the extent to which they experienced positive changes following trauma (0 = I did not experience this change as a result of my crisis– 5 = I experienced this change to a very great degree as a result of my crisis). Scores are summed for a total score and each subscale can be summed for dimensional scores. Items asking about the nature of the trauma and time since it occurred were removed to avoid repetition from the PDS measure. The scale has been validated with clinical samples, has good internal consistency (α = .90, test–retest reliability of .71) and replicable factor structure with the original measure [44]. In the current study, good internal consistency is reported (α = 0.89).

**The expressions of spirituality inventory–revised.** (ESI-R;[45]) is a short 30-item version of the original 100-item measure. Items 31 ("This questionnaire appears to be measuring spirituality") and 32 ("I responded to all statements honestly") were not included in the current study, as they do not contribute to the calculation of final scores. Respondents are asked to the extent to which they agree with 30 statements pertaining to spirituality (0 = strongly disagree—4 = strongly agree). The ESI-R includes five distinct factors or dimensions (of six questions each): Cognitive Orientation toward Spirituality, Experiential/ Phenomenological, Existential Well-Being, Paranormal Beliefs, and Religiousness, which combine to produce a total spirituality score [37]. The ESI-R has been found to demonstrate good face validity, and Cronbach's alpha reliability of the subdivisions was found to range between .72 and .89. In the current study, α = 0.92 indicates very good internal consistency.

## Procedure

Students were recruited via internal advertisement on the homepage of the University website. Participants were asked to confirm consent online and anonymously complete the study measures. Upon completion, participants were directed to an independent link to enter their email address to receive a £5 Amazon voucher to thank them for their time. As the study addressed sensitive issues, signposting information regarding student support services were provided and the researchers were contactable via email.

## Data analysis procedure

**Data cleaning and missing data.** Data were examined for accuracy and 12 cases of duplicate data were removed (identified via timings of data entries matched with timings of multiple applications for vouchers made from email addresses comprised of derivatives of the same username). The percentage of missing data for each measure were as follows: PDS (0.36%), ERRI (0.15%), PTGI-SF (0.20%) and the ESI-R (0.27%). Little's 'missing completely missing at

random' (MCAR) confirmed data were missing at random. The series mean was substituted for missing data in cases where participants had missed 10% or less of items in one measure [46].

**Testing assumptions.** The PDS, PTGI-SF and ERRI totals were found to be significantly not normal despite data transformation and standardisation, therefore non-parametric correlation analyses were used for all measures. Tests for tolerance and variance inflation factors (VIF) did not indicate any evidence of multicollinearity. No evidence of homoscedasticity was detected upon visual inspection of plots. All cases were within acceptable Cook's distance limits. Four cases were found to have standardised residuals outside of the limits of -2 to +2, and one case was found to be above the cut-off point for Mahalanobis, however this is deemed acceptable for the sample size for regression analyses [47].

**Data analysis.** Data were calculated using SPSS (22.0) statistical software. Mean PTGI-SF score of those with and without head injuries, and those who did and did not receive therapy, were compared. No significant difference between the groups was found, therefore neither factor was added as a co-variate. Bivariate correlations were conducted on all variables (point biserial correlations were performed on the nominal correlations that included gender and ethnicity). Bonferroni correction was conducted to indicate an adjustment for the effect of multiple correlations ($p = .002$). Only variables that were found to correlate with the dependent variables of PTG and PTSD were included as co-variates in the final moderation and regression analyses.

Moderation analysis of both PTG and PTSD models was conducted using model one in PROCESS macro for SPSS [48]. The PTG model was found to be significant, therefore, further investigation of simple slopes for the relationship between DR and PTG were tested for low (-1 SD below the mean), moderate (mean), and high (+1 SD above the mean) levels of spirituality. Additionally, stepwise hierarchical regression analyses were performed to assess the predictive power of the variables on PTG. Ethnicity was entered as a co-variate in block one; DR as an independent variable in block two; and spirituality (total score) entered in block three as a moderator. The PTSD moderation model was non-significant, therefore stepwise hierarchical regression analyses were performed, in line with recommendations by Wuensch [49]. IR was entered as an independent variable in the first block and spirituality (total score) was entered in the second block as a moderator.

# Results

## Descriptive statistics

Descriptive results are presented in Table 2 with comparative means where possible. The PDS indicates that the mean number of PTSD symptoms experienced was less than the normative PTSD patient sample, but the subscales were all more than a non-clinical sample. These findings are therefore not representative of a patient sample. Participants experienced less IR, DR and PTG than a comparative sample of American students [50]. Participants experienced the most growth in the domains of 'appreciation of life' and the least in relation to 'spiritual change'. No comparative total is available for the ESI-R, however, participants scored lower on all subscales compared to the normative sample. The maximum score was found in all domains of spirituality except 'paranormal beliefs'.

## Correlations

Bivariate correlation matrix of the main study variables including all subscales is presented in Table 3. Spearman's rho data analysis revealed no significant relationship between TST and PTG. As predicted, a strong positive relationship was found between DR and PTG, suggesting

**Table 2. Mean & SD scores of main variables.**

|  |  | Sample Mean & (SD) | Range | Comparative Mean & (SD) |
|---|---|---|---|---|
| **PDS** | Re-experiencing | 4.69 (3.71) | 0–15 | 1.02 (0.88)[1] |
|  | Avoidance | 6.74 (5.36) | 0–21 | 0.75 (0.74)[1] |
|  | Arousal | 4.97 (4.41) | 0–15 | 0.99 (0.98)[1] |
|  | Total | 16.41 (12.02) | 0–48 | 23.41 (14.68)[2] |
| **ERRI** | Intrusive | 18.78 (8.44) | 1–30 | 15.22 (8.75)[3] |
|  | Deliberate | 14.60 (7.34) | 2–30 | 14.16 (7.88)[3] |
|  | **Total** | 33.38 (13.69) | 4–60 | - |
| **PTGI** | Relating to others | 3.76 (3.03) | 0–10 | 3.04 (1.12)[4] |
|  | Appreciation of life | 5.65 (2.92) | 0–10 | 3.38 (1.27)[4] |
|  | Personal strength | 5.50 (2.93) | 0–10 | 2.86 (1.23)[4] |
|  | Spiritual change | 2.74 (3.19) | 0–10 | 1.39 (1.39)[4] |
|  | New possibilities | 3.72 (3.19) | 0–10 | 1.85 (1.30)[4] |
|  | **Total** | 21.40 (12.16) | 2–50 | 31.84 (18.30)[3] |
| **ESI-R** | Cognitive Orientation | 10.96 (7.48) | 0–24 | 15.21 (5.20)[5] |
|  | Experiential/phenomenological | 7.17 (6.06) | 0–24 | 10.62 (4.91)[5] |
|  | Existential well-being | 12.84 (5.95) | 0–24 | 15.05 (4.30)[5] |
|  | Paranormal beliefs | 8.28 (5.41) | 0–19 | 10.72 (4.68)[5] |
|  | Religiousness | 9.14 (7.55) | 0–24 | 14.64 (5.85)[5] |
|  | **Total** | 48.42 (22.66) | 0–106 | - |

Note

[1] Gul (2014)

[2] Foa (1995)

[3] Lancaster et al. (2015)

[4] Redwood (2015)

[5] MacDonald (2000).

the more engagement in DR, the more growth is also experienced. A moderately significant relationship was found between spirituality (ESI-R) and PTG, and significant correlations were found between spirituality and four subscales of the PTGI. This suggests that the more spiritual one is, the more growth one is likely to demonstrate in almost all domains, with the exception of 'new possibilities'. Furthermore, the PTGI total was significantly correlated with all spirituality subscales, except for existential well-being and paranormal beliefs, suggesting that the more growth one achieves, the more likely one is to score as spiritual in all but two domains.

No significant relationship was found between spirituality and total symptoms of PTSD. The existential well-being subscale of the ESI-R was significantly negatively correlated with the PDS total, all three PDS subscales and IR suggesting that greater levels of existential well-being are associated with fewer PTSD symptoms, and vice versa.

Additionally, a significant positive relationship was found between DR and spirituality. This suggests that greater the spirituality, the more likely they are to engage in DR, and conversely, the more someone engages in DR, the more likely they are to report having spiritual beliefs. Gender was not significantly related with any of the variables. Significant positive relationships were found between ethnicity and total PTG, and the subscale of spiritual change; ethnicity was also positively correlated with total spirituality and the subscales of cognitive orientation and religiosity.

**Table 3. Correlation matrix of all variables including subscales.**

| Variable | 1± | 2. | 3.± | 4. | 5. | 6. | 7. | 8. | 9. | 10. | 11 | 12 | 13 | 14 | 15 | 16 | 17 | 18 | 19 | 20 | 21 | 22 |
|---|---|---|---|---|---|---|---|---|---|---|---|---|---|---|---|---|---|---|---|---|---|---|
| 1. Gender ± | - | | | | | | | | | | | | | | | | | | | | | |
| 2. Age | -.071 | - | | | | | | | | | | | | | | | | | | | | |
| 3. Ethnicity± | -.169 | .001 | - | | | | | | | | | | | | | | | | | | | |
| 4. Time since trauma | -.124 | .339* | .066 | - | | | | | | | | | | | | | | | | | | |
| 5. PDS symptoms total | .061 | -.113 | .143 | -.226 | - | | | | | | | | | | | | | | | | | |
| 6. PDS re-experiencing | .010 | -.079 | .018 | -.296* | .828* | - | | | | | | | | | | | | | | | | |
| 7. PDS avoidance | -.013 | -.167 | .176 | -.163 | .901* | .617* | - | | | | | | | | | | | | | | | |
| 8. PDS arousal | .169 | -.040 | .088 | -.146 | .897* | .689* | .704* | - | | | | | | | | | | | | | | |
| 9. ERRI Total | .178 | .275 | .133 | -.081 | .525* | .479* | .389* | .536* | - | | | | | | | | | | | | | |
| 10. Deliberate rumination | .116 | .232 | .244 | -.077 | .424* | .397* | .319* | .400* | .833* | - | | | | | | | | | | | | |
| 11. Intrusive Rumination | .181 | .198 | -.040 | -.118 | .478* | .436* | .340* | .513* | .863* | .477* | | | | | | | | | | | | |
| 12. PTG Total | .009 | .171 | .336* | .135 | .221 | .143 | .236 | .210 | .460* | .617* | .178 | | | | | | | | | | | |
| 13. PTG: appreciation of life | .028 | .149 | .221 | .039 | .277 | .161 | .313* | .221 | .366* | .470* | .192 | .816* | | | | | | | | | | |
| 14. PTG: relations with others | .119 | .168 | .287 | .190 | -.030 | -.037 | -.034 | -.006 | .290 | .419* | .065 | .791* | .532* | | | | | | | | | |
| 15. PTG: personal strength | .072 | .247 | .161 | .152 | .075 | -.007 | .105 | .097 | .288 | .411* | .084 | .744* | .641* | .469* | | | | | | | | |
| 16. PTG: spiritual change | -.094 | .065 | .471* | .115 | .284 | .225 | .264 | .272 | .494* | .608* | .221 | .752* | .463* | .587* | .373* | | | | | | | |
| 17. PTG: new possibilities | -.046 | .049 | .279 | .061 | .280 | .222 | .312* | .231 | .389* | .574* | .110 | .819* | .599* | .540* | .495* | .609* | | | | | | |
| 18. ESI Total | -.115 | .130 | .468* | .131 | .055 | -.005 | .097 | .052 | .337* | .358* | .200 | .481* | .325* | .406* | .332* | .604* | .297 | | | | | |
| 19. ESI: cognitive orientation | -.046 | .146 | .455* | .177 | .128 | .033 | .164 | .136 | .378* | .419* | .230 | .501* | .411* | .370* | .349* | .624* | .304 | .882* | | | | |
| 20. ESI: phenomenological | -.071 | -.075 | .241 | .009 | .162 | .075 | .142 | .193 | .349* | .322* | .290* | .391* | .227 | .347* | .170 | .528* | .294 | .789* | .670* | | | |
| 21. ESI: paranormal | .000 | -.010 | .239 | -.095 | .158 | .035 | .157 | .167 | .183 | .143 | .166 | .240 | .210 | .203 | .144 | .285 | .139 | .653* | .434* | .596* | | |
| 22. ESI: existential well-being | -.092 | .120 | .119 | .167 | -.471* | -.333* | -.402* | -.463* | -.277 | -.128 | -.387* | .125 | -.033 | .195 | .233 | .076 | .071 | .143 | -.013 | -.091 | -.145 | |
| 23. ESI: religiosity | -.101 | .181 | .489*. | .121 | .097 | .029 | .118 | .104 | .347* | .335* | .231 | .390* | .251 | .374* | .277 | .539* | .170 | .878* | .820* | .594* | .529* | -.025 |

Note: PDS = Post-traumatic Distress Scale, PTGI = Post-traumatic Growth Inventory; ESI = Expressions of Spirituality Inventory

* p < .002

± Point biserial correlations

**Table 4. Linear model of predictors of PTG.**

| | B | SE B | t | p |
|---|---|---|---|---|
| Constant | 19.73 [17.18, 22.29] | 1.286 | 15.34 | p < .001 |
| Spirituality (centred) | 0.15 [0.07, 0.24] | 0.043 | 3.67 | p < .001 |
| Deliberate Rumination (centred) | 0.85 [0.62, 1.09] | 0.118 | 7.22 | p < .001 |
| Spirituality X Deliberate Rumination | 0.014 [-.39, .026] | 0.006 | 2.34 | p = .02 |

## Moderation analysis

A significant interaction (see Table 4) was found between the DR and spirituality ($b = 0.014$, $SE_b = 0.006$, $t = 1.34$, $p = .02$). This finding suggests that spirituality is a moderator in the relationship between DR and PTG. However, the non-significant finding in the PTSD model indicates that spirituality does not moderate the relationship between IR and PTSD. Simple slopes for the relationship between DR and PTG indicated that DR was more strongly related to PTG for those with high ($b = 1.178$, 95% CI [0.85, 1.49], $t = 7.34$, $p < 0.001$) and moderate ($b = 0.8522$, 95% CI [0.62, 1.08], $t = 7.22$, $p < 0.001$) levels of spiritualty than those with low levels ($b = 0.526$, 95% CI [0.12, 0.93], $t = 2.60$, $p = 0.01$) of spirituality. These results indicate that the relationship between DR and PTG becomes stronger in people with average to high levels of spirituality (see Fig 2 for the simple slopes graph).

## Hierarchical regression

A three-stage hierarchical stepwise regression was conducted with PTG as the DV, revealing that at stage one, ethnicity did not significantly and uniquely contribute to the model, $F(1, 94) = 13.268$, $p = .125$, however, it explained 12.4% of the variation in PTG. Introducing DR accounted for an additional 31.7% of the variation in PTG, which was significant, $F(1, 93) = 52.690$, $p < .001$. Adding spirituality explained an extra 7.8% and this was also significant $F(1, 92) = 14.884$, $p < .001$. The strongest predictor of PTG was DR. In total, the variables explained

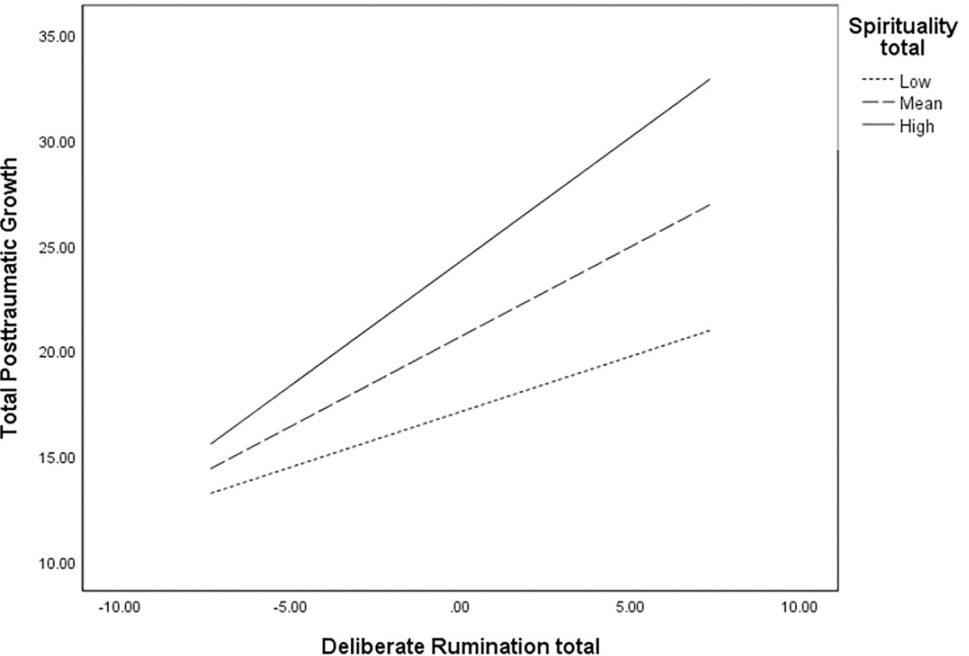

**Fig 2. Simple slopes line graph of levels of spirituality.**

**Table 5. Summary of hierarchical regression models.**

| Model | Step | Variable | $R^2$ | $R^2$ change | Standardized Coefficients | | | 95% Confidence intervals for Beta | |
|---|---|---|---|---|---|---|---|---|---|
| | | | | | Beta | T | Sig | Lower | Upper |
| PTG | Step 1 | Ethnicity | .124 | .124 | .122 | 1.549 | .125 | -.107 | .866 |
| | Step 2 | DR | .441 | .317 | .502 | 6.555 | .000 | .579 | 1.083 |
| | Step 3 | Spirituality | .519 | .078 | .312 | 3.858 | .000 | .081 | .254 |
| PTSD | Step 1 | IR | .241 | .241 | -.491 | 5.464 | .000 | .445 | .954 |

51.9% of the variance in PTG. A further two-stage hierarchical stepwise regression was conducted with PTSD as the dependent variable. The equation removed spirituality as not contributing to the model. Introducing the IR variable accounted for 24.1% of the variation in PTSD, which was significant, $F(1, 94) = 29.86$, $p < .001$ (see Table 5 for the results of both regression models).

## Discussion

As hypothesised, a significant relationship was found between spirituality, PTG and four of the five growth subscales. A significant relationship was found between PTG and all domains of spirituality, except for existential well-being and paranormal beliefs. Spirituality was also found to significantly moderate the relationship between DR and PTG, with the relationship between DR and PTG found to be stronger in people with higher levels of spirituality. However, both of the PTSD-related hypotheses were not supported—no significant relationship was found between spirituality and PTSD, and spirituality did not account for any variance in the PTSD regression model. This suggests that although spirituality is not associated with negative trauma outcomes, it is related to positive trauma outcomes and may play a role in facilitating growth following a traumatic experience. These findings contradict previous findings that higher levels of spirituality are associated with lower levels of distress and lower levels of spirituality are associated with higher levels of distress (e.g., [38]). The moderating relationship reported here provides a more nuanced understanding of this relationship and supports the theory that having a belief system can provide individuals with a framework that facilitates DR and attempts at understanding and meaning-making in the aftermath of distressing life events [33].

The subscale of 'existential well-being' appears to be an anomaly amongst the spirituality subsections. It is the only domain of spirituality that is significantly and negatively associated with all PTSD subscales and IR, and is also one of two domains not related to PTG. In considering the items that comprise the subscale, items are phrased (reverse scored) with more of a focus on the individual alone rather than a broader sense of spirituality ("It always seems that I am doing things wrong", "I am not comfortable with myself", "Much of what I do in life seems strained", "My life is often troublesome", "I often feel tense", and "I am an unhappy person"). Existential well-being may be defined as subjective wellbeing in relation to meaning, purpose and satisfaction in life, as well as comfort in relation to death and suffering. When considering the items listed within this subscale, those reporting greater existential wellbeing report more acceptance and satisfaction with life. This may provide a platform where there is less need to make sense of events that are traumatic. We could consider existential wellbeing to function in a similar way as the construct of resilience (see [45]), which enables coping when faced with distress, but does not necessarily help an individual to grow following a trauma. Resilience is

inversely related to both PTSD and PTG [51]. Resilience has been defined as an individual's ability to maintain equilibrium when experiencing aversive life circumstances [52], whereas PTG is an individual's capacity to use the process of distress to enable them to make improvements in their life following a trauma [43]. Bonanno, Wortman and Nesse [53] propose that resilience provides a stability that results in less struggle with the aftermath of trauma, resulting in less of a need to make sense of events. It may be possible that 'existential well-being' provides a similar function which requires further exploration in the context of trauma.

Consistent with existing literature, a strong positive relationship was found between DR and PTG. DR explained a significant amount of the PTG regression model and was also found to be the strongest predictor of PTG amongst the variables tested, further supporting Calhoun and Tedeschi's [5] theoretical model of DR as an important cognitive process in the development of PTG. The deliberate re-examining and repetitive thinking about beliefs before and after trauma actively rebuilds understanding of an individuals' place in, and assumptions of, a posttraumatic world. Engaging in DR is proposed as a protective factor against PTSD and a predictive factor for PTG [54]. This may be due to the fact that DR, as a deliberate re-examining of the self and the event, can reduce trauma related fear, attenuating the PTSD symptoms and enhancing PTG [54, 55].

Spirituality was found to positively and significantly relate to DR, but not with IR. Overcash and colleagues [33] suggest that metaphysical beliefs are particularly resilient to disconfirmation by empirical evidence; therefore, the association found with DR may be an indication of the cognitive effort required in order to assimilate new evidence into existing belief frameworks. The PTG model proposes that challenges to core beliefs is the causal component that triggers cognitive processing that leads to PTG [3]. Here, DR, actively reconstructing and rebuilding understandings of the self, others, and the world are under investigation and lead to positive post-trauma changes [56]. For those who hold spiritual beliefs as important, being able to use this as a framework from which to explore their core beliefs may be a pathway where an interplay of social, emotional and cognitive processes allow growth the occur. That is, that spirituality may help set DR in motion to process the event and facilitate growth as belief systems can provide frameworks to assimilate challenging life circumstances into one's existing core beliefs [33]. Being supported by a spiritual community may lessen feelings of isolation or grief, forgiveness may be practiced more than revenge, physiological relaxation and reflection may be enhanced through prayer/meditation, and fewer risky behaviours (such as drinking/smoking as coping strategies) may encourage DR in processing the traumatic event.

No significant relationship was found between PTG and TST (as with [28]) and may be a reflection of our limited understanding of the trajectory of PTG over time. No association was found between PTG and gender, contradicting the existing literature (e.g. [23]) that suggests females are more likely to experience PTG. Such gender differences are more apparent with age, in that these differences increase as participants are older [23]. Given our sample were mostly young adults within a small age range, our distribution may be too narrow for such an effect. Furthermore, no relationship was found between PTG and age, in line with Lelorain et al. [22]. Ethnicity was the only co-variate found to correlate with PTG and contribute to the PTG model, but this was not a significant finding. Ethnicity predicting PTG provides some support for Bellizzi et al.'s [19] findings that ethnic minorities are more likely to develop PTG. However, the sample used in this study may not have been ethnically diverse enough for statistical significance.

## Limitations and future research

We note caution in the interpretation of the results. First, we study Type I traumas only, and therefore these findings may not extend to more complex Type II traumas. Second, the use of

cross-sectional, correlational data limits inference of causality and direction of effect, and self-reported retrospective report on cognition in the weeks following a past event may be unreliable. Nevertheless, internal consistency of the measures in the current study are comparable to those reported in other studies.

Third, replicating the study with updated measures would strengthen the findings. The literature on rumination is nuanced, beyond intrusive and deliberate to include reflective rumination and brooding rumination as subtypes of DR [57]. However, measures of reflective and brooding rumination mostly relate to depression rather than in relation to traumatic events. DR and IR in the context of trauma refers specifically to the cognitive processing of an event and its consequences, whereas rumination in the context of depression is more associated with depressive symptoms/mood and its consequences, rather than life events [58, 59]. Adopting measures that explore reflection and perseverative thinking, for example, may further elucidate the relationship of thinking styles in addition to cognitive processing, meaning making, and problem solving for those who have experienced traumatic events. Theoretically, the findings may contribute to our understanding of the nature of both emotional and ruminative processes. Arguments exist over whether the subtypes within these processes (i.e. positive and negative emotions, and deliberate and intrusive rumination) exist at opposite ends of a spectrum or are distinct constructs. Vazquez [60] argues that positive and negative emotions are independent constructs which can co-exist, and Taku, Cann, Tedeschi & Calhoun [61] recommend viewing the subtypes of rumination as multi-dimensional, separate constructs, which may overlap but are essentially distinct. The findings of the current study lend support to the distinct construct theories of both rumination and emotion, as different cognitive variables were found to contribute to both the PTG model (with its associated positive emotions) and PTSD model (with its associated negative emotions). However, Salsman, Segerstrom, Brechting, Carlson and Andrykowski [62, p. 39] state that "much work remains to further delineate the nature of cognitive processing" as the available empirical literature is inconclusive.

The definition of trauma has changed since the publication of the DSM-V [63], however as no measure was found to suitably correlate with the changed definition, the PDS was chosen as it was developed in line with the trauma criteria outlined in DSM-IV [2]. Spirituality, whilst considered distinct from religiosity [35], is a problematic concept to define and operationalise. The ESI-R [45] provides a framework with which to capture the multi-dimensional nature of spirituality. however, it is acknowledged that any measure attempting to capture and operationalise a concept as elusive and individualistic as 'spirituality' is bound to be flawed. Whilst there are inherent difficulties of working empirically with a concept lacking in clarity, it is important that such concepts are not neglected. As Taylor [64] argues, psychology needs to find a scientific method of measuring value systems or it risks losing the human element from which it stemmed, especially in regards to the subject of trauma, when values and beliefs of this kind are often relied on for comfort and strength. Exploring such concepts more fully via qualitative accounts of spirituality post-trauma would be useful to provide perspectives to the proposed frameworks suggested here. For example, meaning-making may act as one explanation for the relationship between DR, spirituality and growth, however, other aspects of spirituality are worth exploring. Positive aspects of belonging to a spiritual community, a sense of mastery and control in accepting unalterable circumstances, or closeness and comfort in having a spiritual connection to a higher power may explain aspects of the experience.

## Clinical implications

Spiritual beliefs may be considered one of a number of mechanisms that can encourage engagement in DR in order to grow and adapt in the aftermath of trauma. The emerging field

of positive clinical psychology argues that encouraging growth would make clinical psychology a more integrative discipline [1]. Clinically the results can assist in the assessment and formulation of individuals with spiritual, not necessarily religious, beliefs who have experienced trauma, and an understanding of PTG may provide clients with a sense of hope in the aftermath of tragedy. Belief systems can be used as a framework and context in which to promote DR in order to derive some benefit from distress [65]. For example, belief systems can serve as a starting point from which to make sense of the world, find meaning in experience, find the positives in distressing experiences and grow and develop psychologically in the aftermath of trauma. Even if clients do not hold spiritual beliefs, therapy could be tailored to encourage particular thinking styles that are associated with spirituality, such as acceptance [66] to aid reflective rumination and foster psychological growth. Additionally, therapies such as Eye-Movement Desensitization and Reprocessing (EMDR) can support the development of DR by facilitating a purposeful exploration of post-traumatic symptoms, supporting the exploration of the mechanisms of the traumatic response in terms of emotion regulation, threat, and memory [67].

## Conclusion

Spirituality was associated with both DR and PTG in the current study and was found to moderate the relationship between the two constructs. However, it was not correlated with PTSD symptoms and did not moderate the relationship between IR and PTSD. This suggests that spirituality is not associated with adverse outcomes of trauma but may help individuals to derive benefit in the aftermath of distressing events.

## Acknowledgments

The authors would like to thank Dr Pierce O'Carroll for his support in the early design of this work, Dr Valentina Lorenzetti and Dr Andy Jones for their support with the analysis, and Dr Gundi Kiemle for reviewing a draft of this manuscript.

## Author Contributions

**Conceptualization:** Catrin Eames, Donna O'Connor.

**Data curation:** Donna O'Connor.

**Methodology:** Catrin Eames.

**Project administration:** Catrin Eames, Donna O'Connor.

**Supervision:** Catrin Eames.

**Writing – original draft:** Catrin Eames, Donna O'Connor.

**Writing – review & editing:** Catrin Eames.

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
