## [Decision Letter · Decision Letter 0]

12 May 2021

PONE-D-21-05935

The role of repetitive thinking and spirituality in the development of posttraumatic growth and symptoms of posttraumatic stress disorder

PLOS ONE

Dear Dr. Earnes,

Thank you for submitting your manuscript to PLOS ONE. After careful consideration, we feel that it has merit but does not fully meet PLOS ONE’s publication criteria as it currently stands. Therefore, we invite you to submit a revised version of the manuscript that addresses the points raised during the review process.

We look forward to receiving your revised manuscript.

Kind regards,

Vedat Sar, M.D.

Academic Editor

PLOS ONE

Journal Requirements:

Reviewers' comments:

Reviewer's Responses to Questions

**Comments to the Author**

1. Is the manuscript technically sound, and do the data support the conclusions?

Reviewer #1: Partly

2. Has the statistical analysis been performed appropriately and rigorously? 

Reviewer #1: I Don't Know

3. Have the authors made all data underlying the findings in their manuscript fully available?

Reviewer #1: Yes

4. Is the manuscript presented in an intelligible fashion and written in standard English?

Reviewer #1: Yes

5. Review Comments to the Author

Reviewer #1: Review (Ken Benau, PhD)

PLOS ONE

YOU MAY POST MY FEEDBACK, BUT I PREFER YOU NOT USE MY NAME.

OVERALL: I RECOMMEND THIS PAPER BE ACCEPTED WITH REVISIONS. SEE BELOW. I BELIEVE THIS RESEARCH IS RELEVANT, BUT THE AUTHORS WOULD BENEFIT FROM A MORE COMPLEX UNDERSTANDING OF THE PSYCHOLOGICAL AND RELATIONAL/ATTACHMENT MECHANISMS THAT CONTRIBUTE TO PTSD AND PTG.

The role of repetitive thinking and spirituality in the development of posttraumatic growth and symptoms of posttraumatic stress disorder

Abstract, and general comment: Why call it “deliberate rumination” rather than “deep, continuous thinking”, “reflective thinking”, or the like. Rumination has a negative connotation, at least for many psychotherapists. It tends to be associated with anxious thinking that is stuck, “rat on the wheel” thinking, that doesn’t get traction and make forward movement. The rumination that comes with PTSD is not the same as “deep, continuous thinking or problem solving”.

p. 2-3: I think the definition of trauma should include not only physical/psychological threat, but mind/body overwhelm, that is more than the person can bear given their consttitution, developmental status, intra-personal and inter-personal resources, etc. at the time of the event. If no overwhelm, no helplessness and powerlessness in the face of the event, then no trauma.

p. 4: While trauma’s impact on cognition and beliefs is very important, it is equally important to recognize the contribution of somatic experience and emotions on the survivor’s post-traumatic experience. I would at minimum reference the somatic and emotional domains of experience, in addition to the cognitive/beliefts.

p. 4: Again, intrusive thoughts are always accompanied by intrusive somatic experience and affect. It would important not to leave out the somatic/affective in this discussion, even though it is not the authors’ research focus.

p. 4-5: While I don’t like the term “deliberative rumination”, I think the key to its distinction from “intrusive rumination” is a) DR’s ability to change rather than remain repetitive and stuck; and 2) the DR person’s capacity for self-reflection. Without the capacity for self-reflection, that is to think about and give new meaning to traumatic experience, there can be no successful processing of the trauma, and therefore no post-traumatic group (ptg).

p. 5: While perhaps not absolutely necessary, I think it would be interesting if the authors’ briefly hypothesize as to why spirituality is associated with lower levels of distress, particularly post-trauma.

p. 6: “1.Controlling for demographic variables, PTSD and IR, it is predicted that TST, DR and spirituality will all show unique positive associations with PTG.

What is “TST”? (Also, on p. 8, p. 14, 21, etc.)

p. 7: “Inclusion criteria: University students were eligible to take part in the survey if they were over 18 years and had experienced a traumatic life event after the age of 16 years. This age limit was chosen in an attempt to capture incidents of Type I trauma (rather than complex Type II trauma), and to ensure that participants had cognitively processed the trauma as adults and not as children.”

I’d recommend the authors explain why they were focusing on Type 1 trauma. Also, please specify what percentage were Type I vs Type II trauma. I would also consider comparing interpersonal vs. non-human (e.g. car accident, natural disaster, illness) trauma, as that could potentially differentiate with respect to PTG and the effects of spirituality.

p. 9: “The Event Related Rumination Inventory (ERRI; [26]) is a 20-item scale used to assess levels of rumination during the weeks immediately following a highly stressful event.”

Since this measure assesses the degree of rumination weeks after the stressful event, how do you know this is an accurate measure 6 months later, for example. The authors should explain how they understand the ERRI is still appropriate 6 months later.

pp. 10-11: With respect to the spirituality measure, the authors should explain if they used the overall score, as contrasted with the subscales, and why, as pertains to their research questions.

p. 17: “Additionally, a significant positive relationship was found between DR and spirituality. This suggests that greater the spirituality, the more likely they are to engage in DR, and conversely, the more someone engages in DR, the more likely they are to report having spiritual beliefs.”

It is important the authors’ hypothesize why they believe there is a significant positive relationship between DR and spirituality. What is the cognitive, emotional, and/or relational, etc. explanation for the correlation.

p. 17: “This finding suggests that spirituality is a moderator in the relationship between DR and PTG.”

It is important the authors’ hypothesize why they believe there is a significant positive relationship between DR and spirituality. What is the cognitive, emotional, and/or relational, etc. explanation for the correlation.

p. 20: “However, the moderating relationship reported here supports the theory that having a belief system can provide individuals with a framework that facilitates DR and attempts at understanding and meaning-making in the aftermath of distressing life events [31].”

While this may be true, the authors’ should think about other ways DR and spirituality contribute to PTG. For example, might these positive correlations have something to do with a self-reflective capacity, seen in both DR and spirituality? Might spirituality allow a person to feel a connection with God/higher power, and this provides a secure attachment to helps the trauma survivor feel less alone, and therefore less likely to feel overwhelmed by trauma reminders. In short, I’d recommend the authors’ consider various psychological variables other than “meaning making” alone that might best explain these correlations. At minimum, I’d recommend the authors’ offer alternative explanations/hypotheses explaining the positive correlations, under the rubric of “Further Study” (pp. 21-22).

p. 20: “The subscale of ‘existential well-being’ appears to be an anomaly amongst the spirituality subsections.”

I’d recommend the authors look at the specific items that make up the “existential well-being” subscale. That might give them greater or less support for their hypothesis. For example, I could imagine “existential” might have more to do with the individual alone, than connected with a higher power. If that were the case, then “alone” vs. “securely attached with an higher power” could significantly impact PTSD and PTG.

p. 20: “Consistent with existing literature, a strong positive relationship was found between DR and PTG.” There may be other explanations to explain this correlation than the one offered by the authors. For example, DR is associated with a capacity for self-reflection, and this capacity to “think about” the self and the traumatic experience could attenuate the PTSD symptoms and enhance PTG. At minimum, I’d like the authors to hypothesize about possible alternative explanations for the DR/PTG correlation, and include this again under “Further Study” (pp. 21-22).

p. 21: “However, measures of reflective and brooding rumination mostly relate to depression rather than in relation to traumatic events.”

I’d recommend the authors reconsider the power of reflection vs brooding as relates to trauma. Reflective vs perseverative thinking is, in my clinical experience, differentially correlated with PTSD and PTG. In short, I would predict the greater reflective capacity, the less PTSD and greater PTG symptoms.

p. 22: “Even if clients do not hold spiritual beliefs, therapy could be tailored to encourage particular thinking styles that are associated with spirituality, such as acceptance [56] to aid reflective rumination and foster psychological growth.”

While thinking styles does impact PTG vs PTSD, in my clinical experience and that of many other, emotional and somatic experience, that is subcortical experience, are more important. The authors should at minimum mention this and recommend it for future research.

6. PLOS authors have the option to publish the peer review history of their article (what does this mean?). If published, this will include your full peer review and any attached files.

Reviewer #1: No

---

## [Author Response · Author response to Decision Letter 0]

10 Nov 2021

Dear Editor

Many thanks for the opportunity to respond to the reviewer comments related to our article “The role of repetitive thinking and spirituality in the development of posttraumatic growth and symptoms of posttraumatic stress disorder”. Please see below each editor/reviewer comment numbered, with our response in bold underneath. Any amendments made are noted in our response here with indicative line numbers, as well as tracked in the manuscript. Any additional amendments made not raised by the reviewers are also noted as well references cited in our response but not included in the manuscript revisions.

Best wishes

Dr Catrin Eames, on behalf of the authors. 

Editor comments

RESPONSE: We have amended the manuscript formatting and file names to adhere to the style requirements. 

RESPONSE: We have checked all references and have reformatted to meet PLOS ONE’s style requirements. Additionally, we have included some further references in some sections that have been added in light of reviewer comments. These are highlighted in the manuscript as track changes. The reference list as a whole is therefore track-changed. 

3. We note that you have indicated that data from this study are available upon request. PLOS only allows data to be available upon request if there are legal or ethical restrictions on sharing data publicly. For more information on unacceptable data access restrictions, please see https://hes32-ctp.trendmicro.com:443/wis/clicktime/v1/query?url=http%3a%2f%2fjournals.plos.org%2fplosone%2fs%2fdata%2davailability%23loc%2dunacceptable%2ddata%2daccess%2drestrictions&umid=3ce35460-db02-4393-9388-706b34028785&auth=768f192bba830b801fed4f40fb360f4d1374fa7c-51078a7fdaf219df28bbd0c56eafb651de7a4837.

b) If there are no restrictions, please upload the minimal anonymized data set necessary to replicate your study findings as either Supporting Information files or to a stable, public repository and provide us with the relevant URLs, DOIs, or accession numbers.

RESPONSE: When submitting the ethics approval for the study, the ethics committee approved access to anonymised data only to those researchers listed in the ethics application and therefore participants did not provide consent for their data to be used in a shared repository. Upon completion of the study, data stored for 5 years then archived. The anonymised dataset is now archived. Contacts for the archived dataset are: recman@liverpool.ac.uk and ethics committee are: ethics@liverpool.ac.uk. 

Reviewers' comments:

Reviewer's Responses to Questions

Comments to the Author

1. Is the manuscript technically sound, and do the data support the conclusions? The manuscript must describe a technically sound piece of scientific research with data that supports the conclusions. Experiments must have been conducted rigorously, with appropriate controls, replication, and sample sizes. The conclusions must be drawn appropriately based on the data presented.

Reviewer #1: Partly

2. Has the statistical analysis been performed appropriately and rigorously?

Reviewer #1: I Don't Know

3. Have the authors made all data underlying the findings in their manuscript fully available?

The PLOS Data policy<https://hes32-ctp.trendmicro.com:443/wis/clicktime/v1/query?url=http%3a%2f%2fwww.plosone.org%2fstatic%2fpolicies.action%23sharing&umid=3ce35460-db02-4393-9388-706b34028785&auth=768f192bba830b801fed4f40fb360f4d1374fa7c-c980035071b360584114d838c3cb5981f33f266b> requires authors to make all data underlying the findings described in their manuscript fully available without restriction, with rare exception (please refer to the Data Availability Statement in the manuscript PDF file). The data should be provided as part of the manuscript or its supporting information, or deposited to a public repository. For example, in addition to summary statistics, the data points behind means, medians and variance measures should be available. If there are restrictions on publicly sharing data—e.g. participant privacy or use of data from a third party—those must be specified.

Reviewer #1: Yes

4. Is the manuscript presented in an intelligible fashion and written in standard English? PLOS ONE does not copyedit accepted manuscripts, so the language in submitted articles must be clear, correct, and unambiguous. Any typographical or grammatical errors should be corrected at revision, so please note any specific errors here.

Reviewer #1: Yes

5. Review Comments to the Author

Reviewer #1: Review 

OVERALL: I RECOMMEND THIS PAPER BE ACCEPTED WITH REVISIONS. SEE BELOW. I BELIEVE THIS RESEARCH IS RELEVANT, BUT THE AUTHORS WOULD BENEFIT FROM A MORE COMPLEX UNDERSTANDING OF THE PSYCHOLOGICAL AND RELATIONAL/ATTACHMENT MECHANISMS THAT CONTRIBUTE TO PTSD AND PTG.

RESPONSE: We are grateful to the reviewer for their helpful comments and recommendation to publish subject to revisions. Our response to each comment made are below:

5.1. 

Abstract, and general comment: Why call it “deliberate rumination” rather than “deep, continuous thinking”, “reflective thinking”, or the like. Rumination has a negative connotation, at least for many psychotherapists. It tends to be associated with anxious thinking that is stuck, “rat on the wheel” thinking, that doesn’t get traction and make forward movement. The rumination that comes with PTSD is not the same as “deep, continuous thinking or problem solving”.

RESPONSE: We acknowledge the reviewer comments in relation to the terminology used, however we would also highlight the inconsistent use of both ‘rumination’ and ‘continuous thinking’ in the literature. Rumination itself may refer to a number of types of recurrent and repetitive thinking, to include reflection and brooding [1], sense making and problem solving [2]. Negative rumination is closely associated with depression and may therefore have negative connotations as highlighted by the reviewer, however, rumination that is related to specific events are not necessarily always negative. Within the context of trauma, active ruminative thinking about an event is theorised to aid the cognitive processing of an event and ways to make sense of it, leading to growth [3-4], whereas some components of rumination about an event can be risk factors for distress. Due to this complexity, a distinction between intrusive and deliberate rumination was proposed [5] to aid understanding of the ruminative process and trauma. Intrusive rumination is not controlled by the individual, focuses on the negative aspects of experience, and can increase distress by thwarting the meaning-making process, whereas deliberate rumination may promote more positive aspects of events as it considers all aspects of the experience. Therefore, deliberate rumination goes beyond “deep continuous thinking” or “reflective thinking” as it is a cognitive process that involves deliberately re-examining an event, that may promote finding meaning in an experience, helping to develop understanding, value and significance from an event [6]. Given these considerations we have chosen to use the terminology ‘deliberate’ and ‘intrusive’ rumination in line with the literature related to post-traumatic growth. We therefore acknowledge the reviewer comments but have not changed the terminology in the manuscript. 

5.2

p. 2-3: I think the definition of trauma should include not only physical/psychological threat, but mind/body overwhelm, that is more than the person can bear given their consttitution, developmental status, intra-personal and inter-personal resources, etc. at the time of the event. If no overwhelm, no helplessness and powerlessness in the face of the event, then no trauma.

RESPONSE: We agree and have expanded in line with the DSM-IV definition of trauma 

Line 36: “…and must elicit a response of intense fear, helplessness or terror”.

5.3

p. 4: While trauma’s impact on cognition and beliefs is very important, it is equally important to recognize the contribution of somatic experience and emotions on the survivor’s post-traumatic experience. I would at minimum reference the somatic and emotional domains of experience, in addition to the cognitive/beliefts.

RESPONSE: We agree with the reviewer that a broader definition in relation to traumatic stress response should be considered and have included the following in the main body of the text 

Line 41-43: “The subjective severity of the event activates emotional distress, and often the distress this causes can prompt individuals to try to make sense of the meaning of the traumatic event(s) to relieve the distress [4, 5].”

5.4

p. 4: Again, intrusive thoughts are always accompanied by intrusive somatic experience and affect. It would important not to leave out the somatic/affective in this discussion, even though it is not the authors’ research focus.

RESPONSE: We have included a clearer definition in line with the PTSD criterion to highlight the reviewer comment, as well as expanding on the definition of IR. 

Line 45-49: “PTSD may be defined by three clusters of symptoms: the re-experiencing of the traumatic event, persistent avoidance of trauma-related stimuli, and heightened persistent symptoms arousal. Memory, affect and response to trauma-stimuli are all effected, with the re-experiencing of the event (e.g., experiencing flashbacks) often with intense emotional experience (e.g., panic) considered a central component of PTSD[2]”.

Line 113-117: IR is a necessary diagnostic criterion for PTSD [29], to include intrusions of persistently re-experiencing via memories, nightmares, flashbacks, emotional distress or physical reactivity after exposure to traumatic reminders [2]. These intrusive thoughts focus on negative aspects and affect of the event and can then prompt the individual to process the trauma in a more conscious and voluntary cognitive manner via somatic cues;

5.5

 p. 4-5: While I don’t like the term “deliberative rumination”, I think the key to its distinction from “intrusive rumination” is a) DR’s ability to change rather than remain repetitive and stuck; and 2) the DR person’s capacity for self-reflection. Without the capacity for self-reflection, that is to think about and give new meaning to traumatic experience, there can be no successful processing of the trauma, and therefore no post-traumatic group (ptg).

RESPONSE: We have defined both IR and DR in line with current accepted definition within the PTG model and feel this is already addressed in the manuscript (definition of IR is expanded as per comment 5.5, DR definition unchanged from line 154). 

5.6

p. 5: While perhaps not absolutely necessary, I think it would be interesting if the authors’ briefly hypothesize as to why spirituality is associated with lower levels of distress, particularly post-trauma.

RESPONSE: We have added a brief note to support what is already articulated about the research related to spirituality and PTG to address this. 

line 177: “…as spirituality can be seen as a source of support and guidance in the face of adversity”

5.7

p. 6: “1.Controlling for demographic variables, PTSD and IR, it is predicted that TST, DR and spirituality will all show unique positive associations with PTG.

What is “TST”? (Also, on p. 8, p. 14, 21, etc.)

RESPONSE: TST is defined on page 4 as ‘time since trauma’ (line 124). Once defined, the acronym is used throughout. We have therefore not defined again on page 8, 14 etc. 

5.8 

p. 7: “Inclusion criteria: University students were eligible to take part in the survey if they were over 18 years and had experienced a traumatic life event after the age of 16 years. This age limit was chosen in an attempt to capture incidents of Type I trauma (rather than complex Type II trauma), and to ensure that participants had cognitively processed the trauma as adults and not as children.”

I’d recommend the authors explain why they were focusing on Type 1 trauma. 

RESPONSE: We have expanded on our consideration of why Type I trauma is more associated with PTG in the introduction 

Line 73-76: “This may be due to the theoretical assumption that events need to be of significance to trigger cognitive rumination, processing, and emotion regulation [6], but not so significant (for example, repeated and prolonged trauma) as to hinder the process [17]”.

And added clarity in the method 

Line 202: “(rather than complex Type II trauma which is less associated with PTG)”

Also, please specify what percentage were Type I vs Type II trauma. 

RESPONSE: We do not have this data, as only those with Type I trauma were included as eligible for the study. Those who met criteria for more complex trauma were excluded at screening. No changes are therefore made in light of this comment.

I would also consider comparing interpersonal vs. non-human (e.g. car accident, natural disaster, illness) trauma, as that could potentially differentiate with respect to PTG and the effects of spirituality.

RESPONSE: We thank the reviewer for this comment, and whilst we acknowledge this may be a useful exploration in the wider literature, we were primarily interested in the cognitive process regardless of the event categorisation in the current manuscript. Given this we only explored ‘potential’ trauma type and did not collect much data in relation to the categorisation of events. For those who did indicate type of trauma, the numbers are relatively small and unequal and therefore would limit the conclusions drawn from subsequent analyses. We therefore have not made any changes in light of this comment. 

5.9

p. 9: “The Event Related Rumination Inventory (ERRI; [26]) is a 20-item scale used to assess levels of rumination during the weeks immediately following a highly stressful event.”

Since this measure assesses the degree of rumination weeks after the stressful event, how do you know this is an accurate measure 6 months later, for example. The authors should explain how they understand the ERRI is still appropriate 6 months later.

RESPONSE: At the time of conducting the study, there was only one questionnaire that measured intrusive and deliberate rumination response to a specific and traumatic event: the Event Related Rumination Inventory [11]. This measure is widely used in the literature. The questionnaire was used in line with measure guidelines, in exploring rumination in the weeks following an event. Although an element of recall could be unreliable, as we state in our discussion (see line 537), the internal consistency in the current study is in line with those reported elsewhere. We have added a statement to reflect this in the limitations section.

Line 538-539: “Nevertheless, internal consistency of the measures in the current study are comparable to those reported in other studies”. 

5.10

pp. 10-11: With respect to the spirituality measure, the authors should explain if they used the overall score, as contrasted with the subscales, and why, as pertains to their research questions.

RESPONSE: Throughout, when referring to ‘spirituality’, the total score is used in line with the hypotheses posed. Where subscales are considered in terms of descriptive explorative data, this has been stated in the text. To clarify we have indicated that ‘total scores’ were used in the data analysis section for the moderator analyses (Lines 336 and 340). 

5.11

p. 17: “Additionally, a significant positive relationship was found between DR and spirituality. This suggests that greater the spirituality, the more likely they are to engage in DR, and conversely, the more someone engages in DR, the more likely they are to report having spiritual beliefs.”

 It is important the authors’ hypothesize why they believe there is a significant positive relationship between DR and spirituality. What is the cognitive, emotional, and/or relational, etc. explanation for the correlation.

RESPONSE: We agree that this should be expanded upon. We have done so in the discussion section, rather than in the results section as suggested here to allow space for greater elaboration. 

Line 505-518: “The PTG model proposes that challenges to core beliefs is the causal component that triggers cognitive processing that leads to PTG [3]. Here, DR, actively reconstructing and rebuilding understandings of the self, others, and the world are under investigation and lead to positive post-trauma changes [56]. For those who hold spiritual beliefs as important, being able to use this as a framework from which to explore their core beliefs may be a pathway where an interplay of social, emotional and cognitive processes allow growth the occur. That is, that spirituality may help set DR in motion to process the event and facilitate growth as belief systems can provide frameworks to assimilate challenging life circumstances into one’s existing core beliefs [33]. Being supported by a social/spiritual community may lessen feelings of isolation or grief, forgiveness may be practiced more than revenge, physiological relaxation and reflection may be enhanced through prayer/meditation, and fewer risky behaviours (such as drinking/smoking as coping strategies) may encourage DR in processing the traumatic event”. 

5.12 

p. 17: “This finding suggests that spirituality is a moderator in the relationship between DR and PTG.”

It is important the authors’ hypothesize why they believe there is a significant positive relationship between DR and spirituality. What is the cognitive, emotional, and/or relational, etc. explanation for the correlation.

RESPONSE: This is addressed as our response to 5.11

5.13

 p. 20: “However, the moderating relationship reported here supports the theory that having a belief system can provide individuals with a framework that facilitates DR and attempts at understanding and meaning-making in the aftermath of distressing life events [31].”

While this may be true, the authors’ should think about other ways DR and spirituality contribute to PTG. For example, might these positive correlations have something to do with a self-reflective capacity, seen in both DR and spirituality? Might spirituality allow a person to feel a connection with God/higher power, and this provides a secure attachment to helps the trauma survivor feel less alone, and therefore less likely to feel overwhelmed by trauma reminders. In short, I’d recommend the authors’ consider various psychological variables other than “meaning making” alone that might best explain these correlations. At minimum, I’d recommend the authors’ offer alternative explanations/hypotheses explaining the positive correlations, under the rubric of “Further Study” (pp. 21-22).

RESPONSE: We have expanded as above, and included further comment in the further study section. 

Line 577-588: “Exploring such concepts more fully via qualitative accounts of spirituality post-trauma would be useful to provide perspectives to the proposed frameworks suggested here. For example, meaning-making may act as one explanation for the relationship between DR, spirituality and growth, however, other aspects of spirituality are worth exploring. Positive aspects of belonging to a spiritual community, a sense of mastery and control in accepting unalterable circumstances, or closeness and comfort in having a spiritual connection to a higher power may explain aspects of the experience”. 

5.14

p. 20: “The subscale of ‘existential well-being’ appears to be an anomaly amongst the spirituality subsections.”

 I’d recommend the authors look at the specific items that make up the “existential well-being” subscale. That might give them greater or less support for their hypothesis. For example, I could imagine “existential” might have more to do with the individual alone, than connected with a higher power. If that were the case, then “alone” vs. “securely attached with an higher power” could significantly impact PTSD and PTG.

RESPONSE: As the reviewer suggests, the subscale is phrased more towards the individual – we have included this to further support our discussion here. 

Line 471-473

In considering the items that comprise the subscale, items are phrased with more of a focus on the individual alone. With that in mind, existential well-being may be akin to resilience 

5.15

p. 20: “Consistent with existing literature, a strong positive relationship was found between DR and PTG.” There may be other explanations to explain this correlation than the one offered by the authors. For example, DR is associated with a capacity for self-reflection, and this capacity to “think about” the self and the traumatic experience could attenuate the PTSD symptoms and enhance PTG. At minimum, I’d like the authors to hypothesize about possible alternative explanations for the DR/PTG correlation, and include this again under “Further Study” (pp. 21-22).

RESPONSE: We have included some further consideration about the relationship in the discussion. 

Line 488-491: ‘Engaging in DR is proposed as a protective factor against PTSD and a predictive factor for PTG [54]. This may be due to the fact that DR, as a deliberate re-examining of the self and the event, can reduce trauma related fear, attenuating the PTSD symptoms and enhancing PTG [54-55].

5.16

p. 21: “However, measures of reflective and brooding rumination mostly relate to depression rather than in relation to traumatic events.”

I’d recommend the authors reconsider the power of reflection vs brooding as relates to trauma. Reflective vs perseverative thinking is, in my clinical experience, differentially correlated with PTSD and PTG. In short, I would predict the greater reflective capacity, the less PTSD and greater PTG symptoms.

RESPONSE: We agree this requires some broader reflection, and have included this in the discussion. 

Line 544-565: “DR and IR in the context of trauma refers specifically to the cognitive processing of an event and its consequences, whereas rumination in the context of depression is more associated with depressive symptoms/mood and its consequences, rather than life events [58-59]. Adopting measures that explore reflection and perseverative thinking, for example, may further elucidate the relationship of thinking styles in addition to cognitive processing, meaning making, and problem solving for those who have experienced traumatic events. 

Theoretically, the findings may contribute to our understanding of the nature of both emotional and ruminative processes. Arguments exist over whether the subtypes within these processes (i.e. positive and negative emotions, and deliberate and intrusive rumination) exist at opposite ends of a spectrum or are distinct constructs. Vazquez [60] argues that positive and negative emotions are independent constructs which can co-exist, and Taku, Cann, Tedeschi & Calhoun [61] recommend viewing the subtypes of rumination as multi-dimensional, separate constructs, which may overlap but are essentially distinct. The findings of the current study lend support to the distinct constructs theories of both rumination and emotion, as different cognitive variables were found to contribute to both the PTG model (with its associated positive emotions) and PTSD model (with its associated negative emotions). However, Salsman, Segerstrom, Brechting, Carlson and Andrykowski [62 p. 39] state that "much work remains to further delineate the nature of cognitive processing” as the available empirical literature is inconclusive”.

5.17

p. 22: “Even if clients do not hold spiritual beliefs, therapy could be tailored to encourage particular thinking styles that are associated with spirituality, such as acceptance [56] to aid reflective rumination and foster psychological growth.”

While thinking styles does impact PTG vs PTSD, in my clinical experience and that of many other, emotional and somatic experience, that is subcortical experience, are more important. The authors should at minimum mention this and recommend it for future research.

RESPONSE: We have added a section to consider further therapies in the clinical implications section. 

Line 501-504: “Additionally, therapies such as Eye-Movement Desensitization and Reprocessing (EMDR) can support the development of DR by facilitating a purposeful exploration of post-traumatic symptoms, supporting the exploration of the mechanisms of the traumatic response in terms of emotion regulation, threat, and memory [67]”.

ADDITIONAL AMENDMENTS

After revision of author guidelines and discussion with the authors, Dr Gundi Kiemle is no longer listed as an author but acknowledged for their contribution. 

REFERENCES (cited in response letter)

1. Treynor W, Gonzalez R, Nolen-Hoeksema S. Rumination reconsidered: psychometric analysis. Cog Ther Res. 2003; 27:247–259

2. Martin LL, Tesser A. Some ruminative thoughts. In Wyer RS Jr, editor. Ruminative thoughts: Advances in Social Cognition (vol 9). Mahwah, NJ: Lawrence Erlbaum Associates. 1996. p 1–47.

3. Tedeschi RG, Calhoun L. Posttraumatic Growth: Future Directions. In: Tedeschi RG, Park CL Calhoun LG (editors). Post-Traumatic Growth: Positive Changes in the Aftermath of Crisis. Mahwah: Lawrence Erlbaum. 1998, p 93-102. 

4. Calhoun, LG, Tedeschi, RG. The Foundations of Posttraumatic Growth: An Expanded Framework. In Calhoun LG, Tedeschi RG, editors. Handbook of Posttraumatic Growth, Research and Practice. Mahwah, NJ: Lawrence Erlbaum Associates. 2006. p 3-23.

5. Cann A, Calhoun LG, Tedeschi RG, Taku K, Vishnevsky T, Triplett KN, Danhauer SC. A short form of the posttraumatic growth inventory. Anxiety Stress Coping. 2010;23:127–137. doi: 10.1080/10615800903094273

6. Cann A, Calhoun LG, Tedeschi RG, Triplett KN, Vishnevsky T, Lindstrom CM Assessing posttraumatic cognitive processes: the event related rumination inventory. Anxiety Stress Coping. 2011; 24:137–156

7. Kamijo N, Yukawa S. The Role of Rumination and Negative Affect in Meaning Making Following Stressful Experiences in a Japanese Sample. Front Psychol. 2018;9:2404. https://doi.org/10.3389/fpsyg.2018.02404

 6. PLOS authors have the option to publish the peer review history of their article (what does this mean?<https://hes32-ctp.trendmicro.com:443/wis/clicktime/v1/query?url=https%3a%2f%2fjournals.plos.org%2fplosone%2fs%2feditorial%2dand%2dpeer%2dreview%2dprocess%23loc%2dpeer%2dreview%2dhistory&umid=3ce35460-db02-4393-9388-706b34028785&auth=768f192bba830b801fed4f40fb360f4d1374fa7c-44600eff4c732db3640dd20ed070ffec39a098d0>). If published, this will include your full peer review and any attached files.

Do you want your identity to be public for this peer review? For information about this choice, including consent withdrawal, please see our Privacy Policy<https://hes32-ctp.trendmicro.com:443/wis/clicktime/v1/query?url=https%3a%2f%2fwww.plos.org%2fprivacy%2dpolicy&umid=3ce35460-db02-4393-9388-706b34028785&auth=768f192bba830b801fed4f40fb360f4d1374fa7c-82b8a28593e17ad43fac8ad5b105702aa08c0a95>.

Reviewer #1: No

While revising your submission, please upload your figure files to the Preflight Analysis and Conversion Engine (PACE) digital diagnostic tool, https://hes32-ctp.trendmicro.com:443/wis/clicktime/v1/query?url=https%3a%2f%2fpacev2.apexcovantage.com&umid=3ce35460-db02-4393-9388-706b34028785&auth=768f192bba830b801fed4f40fb360f4d1374fa7c-2a6215b1b14cbdc9aa27a4f955b6413c5f5d1a17. PACE helps ensure that figures meet PLOS requirements. To use PACE, you must first register as a user. Registration is free. Then, login and navigate to the UPLOAD tab, where you will find detailed instructions on how to use the tool. If you encounter any issues or have any questions when using PACE, please email PLOS at figures@plos.org<mailto:figures@plos.org>. Please note that Supporting Information files do not need this step.

 In compliance with data protection regulations, you may request that we remove your personal registration details at any time. (Remove my information/details)<https://hes32-ctp.trendmicro.com:443/wis/clicktime/v1/query?url=https%3a%2f%2fwww.editorialmanager.com%2fpone%2flogin.asp%3fa%3dr&umid=3ce35460-db02-4393-9388-706b34028785&auth=768f192bba830b801fed4f40fb360f4d1374fa7c-ec58063f735a6c39d670c78704c496829887dd5b>. Please contact the publication office if you have any questions.

---

## [Decision Letter · Decision Letter 1]

29 Nov 2021

PONE-D-21-05935R1The role of repetitive thinking and spirituality in the development of posttraumatic growth and symptoms of posttraumatic stress disorderPLOS ONE

Dear Dr. Eames,

Thank you for submitting your manuscript to PLOS ONE. After careful consideration, we feel that it has merit but does not fully meet PLOS ONE’s publication criteria as it currently stands. Therefore, we invite you to submit a revised version of the manuscript that addresses the points raised during the review process.

We look forward to receiving your revised manuscript.

Kind regards,

Vedat Sar, M.D.

Academic Editor

PLOS ONE

Reviewers' comments:

Reviewer's Responses to Questions

**Comments to the Author**

1. If the authors have adequately addressed your comments raised in a previous round of review and you feel that this manuscript is now acceptable for publication, you may indicate that here to bypass the “Comments to the Author” section, enter your conflict of interest statement in the “Confidential to Editor” section, and submit your "Accept" recommendation.

Reviewer #1: (No Response)

2. Is the manuscript technically sound, and do the data support the conclusions?

Reviewer #1: Partly

3. Has the statistical analysis been performed appropriately and rigorously? 

Reviewer #1: I Don't Know

4. Have the authors made all data underlying the findings in their manuscript fully available?

Reviewer #1: Yes

5. Is the manuscript presented in an intelligible fashion and written in standard English?

Reviewer #1: Yes

6. Review Comments to the Author

Reviewer #1: Review of Revised Version:

PLOS ONE

The role of repetitive thinking and spirituality in the development of posttraumatic growth and symptoms of posttraumatic stress disorder --Manuscript Draft-- Manuscript Number: PONE-D-21-05935

Page 1

“The Posttraumatic stress [change to Stress] Diagnostic Scale…

Page 2

“These findings indicate that, although spirituality…”. Suggest you change to, “These findings indicate that while spirituality…”

Page 3

You wrote:

“Often the distress this causes can prompt individuals to try to make sense of the meaning of the traumatic event(s) [4] and failure to find meaning has been found to be associated with poor psychological outcomes, including depression and post-traumatic stress disorder (PTSD) [5].”

Suggest rewrite, to aid in clarity:

“Often the distress this causes can prompt individuals to try to make sense of the meaning of the traumatic event(s) [4]. Failure to find meaning has been found to be associated with poor psychological outcomes, including depression and post-traumatic stress disorder (PTSD) [5].”

…

You wrote:

“In contrast, finding meaning in the aftermath of trauma has the potential to experience profound positive psychological transformation and higher levels of psychological functioning than prior to the event(s).”

Suggest rewrite, for better word usage:

“In contrast, when people find meaning in the aftermath of trauma, they have the potential to experience profound positive psychological transformation and higher levels of psychological functioning than prior to the event(s).”

…

You wrote:

“PTG has been associated more with incidents of Type I trauma, rather than Type II [11] occurring following a wide range of Type I traumatic events, e.g. natural disasters [12]road traffic accidents [13] illnesses [14], and interpersonal violence [15].”

Recommend rewrite for clarity:

“PTG has been associated more with incidents of Type I trauma than Type II [11]. PTG has been shown to occur following a wide range of Type I traumatic events, including natural disasters [12], road traffic accidents [13], illnesses [14], and interpersonal violence [15].”

…

Page 4

You wrote, “…participants from the USA reporting higher levels of PTG than other countries, possibly due to social desirability of responding to challenges with positivity [16].”

Maybe this has something to do with the social, emotional, and financial resource to the survivor. Could add to your remark, above.

…

You wrote:

“…however most agree that it takes time, sometimes many years, to develop [25].”

For clarity, I would change to:

“… however most agree that it takes time, sometimes many years, for PTG to develop [25].”

…

“…‘deliberate rumination’ (DR, [26]).”

I know you are not going to change this term, but I think it is an unfortunate one. “Rumination” is thinking usually association with thinking, like worry, that goes over and over the same thing and does not make forward progress. “Reflection” would be a much better term, and closer to what most needs to happen. It leads to positive change, growth, transformation, etc. For example, see this:

https://www.google.com/search?q=reflection+and+rumination&rlz=1C5CHFA_enUS806US806&oq=reflection&aqs=chrome.3.69i57j35i39j0i67i433j0i67l2j0i67i433j0i20i263i433i512j0i67j46i131i433i512j0i67.5083j0j9&sourceid=chrome&ie=UTF-8

“Self-reflection is a tool that helps us approach life with a growth mindset. ... Where self-reflection is purposefully processing (thinking about) our experiences with the intent of learning something, rumination is when we think over and over about something in the past or future with negative emotions directly linked.”

I would recommend you indicate that you are using the term others have used, but that the DR would be better termed, “deliberate reflection”.

Page 5

You wrote: “Belief systems can provide frameworks that assimilate challenging life circumstances into one’s existing core beliefs [31].”

I recommend you change the word “assimilate” to “accommodate”. See this:

https://www.google.com/search?q=assimilation+versus+accommodation&rlz=1C5CHFA_enUS806US806&oq=assimiliate+versus+accom&aqs=chrome.1.69i57j0i22i30j0i10i22i30j0i390.13783j1j9&sourceid=chrome&ie=UTF-8

Assimilation is the process of using or transforming the environment so that it can be placed in preexisting cognitive structures. Accomodation is the process of changing cognitive structures in order to accept something from the environment.

My view comes from Piaget’s discussion of assimilation and accommodation.

You wrote: “Studies report mixed results in terms of its benefits,…”

Please make more clear what “its” refers to.

…

You wrote: “MacDonald et al. [35] expand on this definition, to include: "spirituality is a natural aspect of human functioning…”

This would read more clear: “MacDonald et al. [35] expand on this definition: "Spirituality is a natural aspect of human functioning…”

GENERAL COMMENT: Do not italicize quotes without indicating those are yours, by writing at end of quote-- (my emphasis).

…

Use comma not semi-colon in this sentence, as follows:

“Spirituality has been found to be a moderator of distress, with higher levels of spirituality associated with lower levels of distress, and lower levels of spirituality associated with higher levels of distress [36].”

…

Recommend rewording this sentence as follows:

“This study objectives explores mechanisms that can promote adaptive outcomes following exposure to traumatic events to inform clinical practice. The interactive relationships of spirituality with the subtypes of repetitive thinking in relation to PTG and PTSD are also explored.”

…

Page 6

Recommend rewording as follows, for greater clarity:

“The current study aims to test if evaluates whether 1) high levels of DR and high levels of spirituality are associated with PTG, and whether 2) spirituality modifies this relationship; and 3) as well as exploring whether the relationship between IR and PTSD is moderated by spirituality.”

…

Before you go to your hypotheses, I think it is important to explain, at least briefly, why you think spirituality and DR would be associated with PTG. Also, your abbreviations are easily forgotten, so I recommend you write the full word with the abbreviation when you reintroduce them, as you do here:

“Hypotheses Relevant to PTG:”

For example, I didn’t know what “TST” refered to, in this sentence, and had to look it up because you mention it only once before, I believe:

“Controlling for demographic variables, PTSD and IR, it is predicted that TST, DR and spirituality will all show unique positive associations with PTG.”

You wrote: “This age limit was chosen in an attempt to capture incidents of Type I trauma (rather than complex Type II trauma), and to ensure that participants had cognitively processed the trauma as adults and not as children.”

Studying Type 1 versus Type 2 is a significant choice, as DR and spirituality may not have the same impact on PTG. I would definitely explain your reasoning for your selective criteria, some.

…

Page 9

You wrote: “Two additional questions were included to ascertain whether they experienced a head injury during the event, and if they received any therapy in relation to the trauma as potential confounders in considering cognitive response to trauma.”

These additional questions are very valuable. Please explain your rationale for their inclusion, i.e., why they are important vis a vis your hypotheses.

…

Likewise, you wrote: “Item 39 (“How long have you experienced the problems that you reported above?”) and part 4 (“Indicate if the problems have interfered with any of the following areas of your life”) of the PDS were not included.”

Please indicate why you excluded these questions, as they look like they might be relevant to your hypotheses.

…

Page 10

I think you need quotation marks for the test items, as I corrected below:

Items 31 (“This questionnaire appears to be measuring spirituality”) and 32 (“I responded to all statements honestly”) were not included in the current study, as they do not contribute to the calculation of final scores.

…

Page 13

You wrote: “The PDS indicates that the mean number of PTSD symptoms experienced was less than the normative PTSD patient sample, but the subscales were all more than a non-clinical sample.”

I haven’t read your discussion, yet, but it is important you make clear the PTSD symptoms you are assessing are not severe, i.e., comparable to most PTSD patients. Therefore, please stipulate that these findings do not necessarily apply to PTSD patients, which would need to be further evaluated.

…

Page 17

You wrote: “A moderate positive relationship was found between IR and symptoms of PTSD, suggesting that the more intrusive thoughts experienced, the more likely symptoms of PTSD are experienced.”

I’m not sure you want to include this, as the definition of PTSD includes intrusive thoughts.

…

You wrote: “A significant interaction (see Table 4) was found between the DR and spirituality (b=0.014, SEb=0.006, t=1.34, p=.02). This finding suggests that spirituality is a moderator in the relationship between DR and PTG.”

Why does this not suggest DR is the moderator between spirituality and PTG? This would help me to better understand your interpretation of this finding.

…

Page 18

You wrote: “…that the relationship between DR and PTG becomes stronger in people with average to high levels of spirituality (see Fig. 2 for the simple slopes graph).”

What is the correlation between DR and spirituality? Is there much overlap? I don’t recall if you assessed that. It would be important to do so, as you want to be sure your assessing distinct factors.

…

Page 19

You wrote: “This suggests that although spirituality is not associated with negative trauma outcomes, it is related to positive trauma outcomes and may play a role in facilitating growth following a traumatic experience. These findings contradict previous findings that higher levels of spirituality are associated with lower levels of distress and lower levels of spirituality are associated with higher levels of distress (e.g., [36]).”

Since your finding differs significantly from previous findings, i.e., positive vs negative effects of spirituality and PTG, it is important you provide a rationale for your distinct findings.

…

Page 20

You wrote: “Existential well-being may be akin to resilience (see [42]), which enables coping when faced with distress, but does not help an individual to grow following a trauma.”

This feels like a stretch to me. At minimum, you would have to explain what statements correspond with “existential well-being”, and what descriptors are used to describe “resilience”, so I better understand how these two factors might be correlated. As it is written here, I am not finding your suggestion compelling.

…

You wrote: “DR explained a significant amount of the PTG regression model and was also found to be the strongest predictor of PTG amongst the variables tested, further supporting Calhoun and Tedeschi’s [51] theoretical model of DR as an important cognitive process in the development of PTG.”

I think this is, perhaps, THE most important finding of your study. You need to explain the relationship, in your view, between DR and PTG. That is, what exactly is the cognitive process of DR that facilitates PTG. Very important, in my view.

…

You wrote: “As predicted, IR was positively associated with symptoms of PTSD, with IR accounting for almost a quarter of the variance in PTSD, supporting the evidence for these processes as involved in processing traumatic experience.”

As I noted above, this hypothesis states that a symptom of PTSD, IR, is a symptom of PTSD. I do not see the value of this hypothesis and finding.

…

Page 21

You wrote: “No association was found between PTG and gender, contradicting the existing literature (e.g. [21]) that suggests females are more likely to experience PTG.”

Again, since this finding is different than previous findings, it would be helpful for you to briefly explain why this is so.

…

Page 22

Typo, line 472. Should be capitalized: “However, “

…

Under Limitations of this study, I think it is very important that you specify that you are studying Type 1 (simple trauma) and not Type 2 (complex trauma). I believe type 2 trauma is what most trauma therapists see and treat, more often.

…

You wrote: “As Taylor[54] argues, psychology needs to find a scientific method of measuring value systems or it risks losing the human element from which it stemmed, especially in regards to the subject of trauma, when values and beliefs of this kind are often relied on for comfort and strength.”

This is a very important statement. Well done.

…

I think several of your clinical implications are of value. However, you wrote:

“Belief systems can be used as a framework and context in which to promote DR in order to derive some benefit from distress [55].”

You would need to flesh this out much more if you are to make this statement. Please explain how “belief systems” “promote DR”. At least offer one way this may be so, and indicate this can be further researched.

7. PLOS authors have the option to publish the peer review history of their article (what does this mean?). If published, this will include your full peer review and any attached files.

Reviewer #1: No

---

## [Author Response · Author response to Decision Letter 1]

3 Jan 2022

Reviewer comments 

1. Page 1

“The Posttraumatic stress [change to Stress] Diagnostic Scale…

RESPONSE: Amended, line 8.

2. Page 2

“These findings indicate that, although spirituality…”. Suggest you change to, “These findings indicate that while spirituality…”

RESPONSE: Amended, line 14.

3. Page 3

You wrote:

“Often the distress this causes can prompt individuals to try to make sense of the meaning of the traumatic event(s) [4] and failure to find meaning has been found to be associated with poor psychological outcomes, including depression and post-traumatic stress disorder (PTSD) [5].”

Suggest rewrite, to aid in clarity:

“Often the distress this causes can prompt individuals to try to make sense of the meaning of the traumatic event(s) [4]. Failure to find meaning has been found to be associated with poor psychological outcomes, including depression and post-traumatic stress disorder (PTSD) [5].”

RESPONSE: Amended, lines 41-43 (note phrasing in the manuscript differed somewhat to what the reviewer notes here). Amended in line with reviewer suggestion. 

4. You wrote:

“In contrast, finding meaning in the aftermath of trauma has the potential to experience profound positive psychological transformation and higher levels of psychological functioning than prior to the event(s).”

Suggest rewrite, for better word usage:

“In contrast, when people find meaning in the aftermath of trauma, they have the potential to experience profound positive psychological transformation and higher levels of psychological functioning than prior to the event(s).”

RESPONSE: Amended, line 53. 

5. You wrote:

“PTG has been associated more with incidents of Type I trauma, rather than Type II [11] occurring following a wide range of Type I traumatic events, e.g. natural disasters [12]road traffic accidents [13] illnesses [14], and interpersonal violence [15].”

Recommend rewrite for clarity:

“PTG has been associated more with incidents of Type I trauma than Type II [11]. PTG has been shown to occur following a wide range of Type I traumatic events, including natural disasters [12], road traffic accidents [13], illnesses [14], and interpersonal violence [15].”

RESPONSE: Amended lines 70-71 (note citation numbers differ in reviewer comments to that noted in the manuscript).

6. Page 4

You wrote, “…participants from the USA reporting higher levels of PTG than other countries, possibly due to social desirability of responding to challenges with positivity [16].”

Maybe this has something to do with the social, emotional, and financial resource to the survivor. Could add to your remark, above.

RESPONSE: Amended, lines 78-83. 

For example, the social pressure to report growth from adversity is greater in trauma survivors from the USA than those in Europe, and may therefore be a cultural expectation [18]. This is noted regardless of social factors such as non-Caucasian ethnicity and lack of education, both of which are associated with higher growth and are risk factors for PTSD, whereas lack of social support is associated with less growth [16]. 

7. You wrote:

“…however most agree that it takes time, sometimes many years, to develop [25].”

For clarity, I would change to:

“… however most agree that it takes time, sometimes many years, for PTG to develop [25].”

RESPONSE: Amended, line 95. 

8. “…‘deliberate rumination’ (DR, [26]).”

I know you are not going to change this term, but I think it is an unfortunate one. “Rumination” is thinking usually association with thinking, like worry, that goes over and over the same thing and does not make forward progress. “Reflection” would be a much better term, and closer to what most needs to happen. It leads to positive change, growth, transformation, etc. For example, see this:

https://www.google.com/search?q=reflection+and+rumination&rlz=1C5CHFA_enUS806US806&oq=reflection&aqs=chrome.3.69i57j35i39j0i67i433j0i67l2j0i67i433j0i20i263i433i512j0i67j46i131i433i512j0i67.5083j0j9&sourceid=chrome&ie=UTF-8

“Self-reflection is a tool that helps us approach life with a growth mindset. ... Where self-reflection is purposefully processing (thinking about) our experiences with the intent of learning something, rumination is when we think over and over about something in the past or future with negative emotions directly linked.”

I would recommend you indicate that you are using the term others have used, but that the DR would be better termed, “deliberate reflection”.

RESPONSE: We have included the following to reflect the reviewer comments here. Lines 116-121. 

In line with the literature related to PTG, we adopt the terminology of deliberate and intrusive rumination throughout the manuscript. Here, deliberate rumination is defined as a cognitive process that involves deliberately re-examining an event, that may promote finding meaning in an experience, helping to develop understanding, value and significance from an event. We acknowledge that deliberate reflection, the process of purposefully thinking about an experience with an intent to learn, can also be applied within this context. 

9. Page 5

You wrote: “Belief systems can provide frameworks that assimilate challenging life circumstances into one’s existing core beliefs [31].”

I recommend you change the word “assimilate” to “accommodate”. See this:

https://www.google.com/search?q=assimilation+versus+accommodation&rlz=1C5CHFA_enUS806US806&oq=assimiliate+versus+accom&aqs=chrome.1.69i57j0i22i30j0i10i22i30j0i390.13783j1j9&sourceid=chrome&ie=UTF-8

Assimilation is the process of using or transforming the environment so that it can be placed in preexisting cognitive structures. Accomodation is the process of changing cognitive structures in order to accept something from the environment.

My view comes from Piaget’s discussion of assimilation and accommodation.

RESPONSE: Amended, line 127.

10. You wrote: “Studies report mixed results in terms of its benefits,…”

Please make more clear what “its” refers to.

RESPONSE: Amended line 128– now reads “Studies report mixed results, with belief systems associated with both positive and negative health outcomes”

11. You wrote: “MacDonald et al. [35] expand on this definition, to include: "spirituality is a natural aspect of human functioning…”

This would read more clear: “MacDonald et al. [35] expand on this definition: "Spirituality is a natural aspect of human functioning…”

RESPONSE: Amended, 133.

12. GENERAL COMMENT: Do not italicize quotes without indicating those are yours, by writing at end of quote-- (my emphasis).

RESPONSE: Amended – quotes now not italicized, lines 132, 133-138.

13. Use comma not semi-colon in this sentence, as follows:

“Spirituality has been found to be a moderator of distress, with higher levels of spirituality associated with lower levels of distress, and lower levels of spirituality associated with higher levels of distress [36].”

RESPONSE: Amended, line 139.

14. Recommend rewording this sentence as follows:

“This study objectives explores mechanisms that can promote adaptive outcomes following exposure to traumatic events to inform clinical practice. The interactive relationships of spirituality with the subtypes of repetitive thinking in relation to PTG and PTSD are also explored.”

RESPONSE: Amended, lines 149-151.

15. Page 6

Recommend rewording as follows, for greater clarity:

“The current study aims to test if evaluates whether 1) high levels of DR and high levels of spirituality are associated with PTG, and whether 2) spirituality modifies this relationship; and 3) as well as exploring whether the relationship between IR and PTSD is moderated by spirituality.” 

RESPONSE: Amended, lines 152-153.

16. Before you go to your hypotheses, I think it is important to explain, at least briefly, why you think spirituality and DR would be associated with PTG. 

RESPONSE: Amended, lines 154-156. 

For those who hold spiritual beliefs as important, spirituality may provide a framework from which growth may occur via an interplay of cognitive, emotional, and social processes to explore core beliefs and challenging life circumstances. 

17. Also, your abbreviations are easily forgotten, so I recommend you write the full word with the abbreviation when you reintroduce them, as you do here:

“Hypotheses Relevant to PTG:”

For example, I didn’t know what “TST” refered to, in this sentence, and had to look it up because you mention it only once before, I believe:

“Controlling for demographic variables, PTSD and IR, it is predicted that TST, DR and spirituality will all show unique positive associations with PTG.”

RESPONSE: Amended, lines 160-163.

18. You wrote: “This age limit was chosen in an attempt to capture incidents of Type I trauma (rather than complex Type II trauma), and to ensure that participants had cognitively processed the trauma as adults and not as children.”

Studying Type 1 versus Type 2 is a significant choice, as DR and spirituality may not have the same impact on PTG. I would definitely explain your reasoning for your selective criteria, some.

RESPONSE: This was already expanded upon in the previous revision, stating that Type II trauma is less associated with PTG – see highlight lines 193-194. No further amendment made. 

19. Page 9

You wrote: “Two additional questions were included to ascertain whether they experienced a head injury during the event, and if they received any therapy in relation to the trauma as potential confounders in considering cognitive response to trauma.”

These additional questions are very valuable. Please explain your rationale for their inclusion, i.e., why they are important vis a vis your hypotheses.

RESPONSE: Amended to include the following text, lines 226-228. 

Two additional questions were included to ascertain whether they experienced a head injury during the event, and if they received any therapy in relation to the trauma. These were included as both factors were deemed important to include as potential confounders when measuring cognitive response to trauma.

…

20. Likewise, you wrote: “Item 39 (“How long have you experienced the problems that you reported above?”) and part 4 (“Indicate if the problems have interfered with any of the following areas of your life”) of the PDS were not included.”

Please indicate why you excluded these questions, as they look like they might be relevant to your hypotheses.

RESPONSE: Amendment, lines 235-283. 

Due to experimental error, item 39 (“How long have you experienced the problems that you reported above?”) and part 4 (“Indicate if the problems have interfered with any of the following areas of your life”) of the PDS were missing from the final online survey and therefore were not included.

21. Page 10

I think you need quotation marks for the test items, as I corrected below:

Items 31 (“This questionnaire appears to be measuring spirituality”) and 32 (“I responded to all statements honestly”) were not included in the current study, as they do not contribute to the calculation of final scores.

RESPONSE: Amended, line 264-265.

22. Page 13

You wrote: “The PDS indicates that the mean number of PTSD symptoms experienced was less than the normative PTSD patient sample, but the subscales were all more than a non-clinical sample.”

I haven’t read your discussion, yet, but it is important you make clear the PTSD symptoms you are assessing are not severe, i.e., comparable to most PTSD patients. Therefore, please stipulate that these findings do not necessarily apply to PTSD patients, which would need to be further evaluated.

RESPONSE: Amended, line 332: 

These findings are therefore not representative of a patient sample.

23. Page 17

You wrote: “A moderate positive relationship was found between IR and symptoms of PTSD, suggesting that the more intrusive thoughts experienced, the more likely symptoms of PTSD are experienced.”

I’m not sure you want to include this, as the definition of PTSD includes intrusive thoughts.

RESPONSE: Amendment : removed in line with reviewer suggestion.

24. You wrote: “A significant interaction (see Table 4) was found between the DR and spirituality (b=0.014, SEb=0.006, t=1.34, p=.02). This finding suggests that spirituality is a moderator in the relationship between DR and PTG.”

Why does this not suggest DR is the moderator between spirituality and PTG? This would help me to better understand your interpretation of this finding.

RESPONSE: No amendment made as we feel we have introduced the analysis throughout the manuscript. To clarify further, the moderation analysis was conducted to test the hypothesis that spirituality would moderate the relationship between DR and PTG. Thus, spirituality was entered as a moderator variable. The simple slopes analysis indicates how the relationship between DR and PTG changes at different levels of spirituality (see Fig. 2). These results reveal that the relationship between DR and PTG becomes stronger as levels of spirituality increase, however, even when DR is low, levels of PTG are still higher in those with higher levels of spirituality compared to those with lower levels (see Fig. 2).

25. Page 18

You wrote: “…that the relationship between DR and PTG becomes stronger in people with average to high levels of spirituality (see Fig. 2 for the simple slopes graph).”

What is the correlation between DR and spirituality? Is there much overlap? I don’t recall if you assessed that. It would be important to do so, as you want to be sure your assessing distinct factors.

RESPONSE: This is reported in Table 1 as .36, which indicate a small to moderate relationship. No amendment made.

26. Page 19

You wrote: “This suggests that although spirituality is not associated with negative trauma outcomes, it is related to positive trauma outcomes and may play a role in facilitating growth following a traumatic experience. These findings contradict previous findings that higher levels of spirituality are associated with lower levels of distress and lower levels of spirituality are associated with higher levels of distress (e.g., [36]).”

Since your finding differs significantly from previous findings, i.e., positive vs negative effects of spirituality and PTG, it is important you provide a rationale for your distinct findings.

RESPONSE: Amendment: line 443-445 These findings contradict previous findings that higher levels of spirituality are associated with lower levels of distress and lower levels of spirituality are associated with higher levels of distress (e.g., [38]). The moderating relationship reported here provides a more nuanced understanding of this relationship and supports the theory that having a belief system can provide individuals with a framework that facilitates DR and attempts at understanding and meaning-making in the aftermath of distressing life events [33].

27. Page 20

You wrote: “Existential well-being may be akin to resilience (see [42]), which enables coping when faced with distress, but does not help an individual to grow following a trauma.”

This feels like a stretch to me. At minimum, you would have to explain what statements correspond with “existential well-being”, and what descriptors are used to describe “resilience”, so I better understand how these two factors might be correlated. As it is written here, I am not finding your suggestion compelling.

RESPONSE: Amendment lines 455-473. In considering the items that comprise the subscale, items are phrased (reverse scored) with more of a focus on the individual alone rather than a broader sense of spirituality (“It always seems that I am doing things wrong”, “I am not comfortable with myself”, “Much of what I do in life seems strained”, “My life is often troublesome”, “I often feel tense”, and “I am an unhappy person”. Existential well-being may be defined as subjective wellbeing in relation to meaning, purpose and satisfaction in life, as well as comfort in relation to death and suffering. When considering the items listed within this subscale, those reporting greater existential wellbeing report more acceptance and satisfaction with life. This may provide a platform where there is less need to make sense of events that are traumatic. We could consider existential wellbeing to function in a similar way as the construct of resilience (see [45]), which enables coping when faced with distress, but does not necessarily help an individual to grow following a trauma. Resilience is inversely related to both PTSD and PTG [51]. Resilience has been defined as an individual’s ability to maintain equilibrium when experiencing aversive life circumstances [52], whereas PTG is an individual’s capacity to use the process of distress to enable them to make improvements in their life following a trauma [43]. Bonanno, Wortman and Nesse [53] propose that resilience provides a stability that results in less struggle with the aftermath of trauma, resulting in less of a need to make sense of events. It may be possible that ‘existential well-being’ provides a similar function which requires further exploration in the context of trauma. 

28. You wrote: “DR explained a significant amount of the PTG regression model and was also found to be the strongest predictor of PTG amongst the variables tested, further supporting Calhoun and Tedeschi’s [51] theoretical model of DR as an important cognitive process in the development of PTG.”

I think this is, perhaps, THE most important finding of your study. You need to explain the relationship, in your view, between DR and PTG. That is, what exactly is the cognitive process of DR that facilitates PTG. Very important, in my view

RESPONSE: Amended line 489-491.

The deliberate re-examining and repetitive thinking about beliefs before and after trauma actively rebuilds understanding of an individuals’ place in, and assumptions of, a posttraumatic world.

29. You wrote: “As predicted, IR was positively associated with symptoms of PTSD, with IR accounting for almost a quarter of the variance in PTSD, supporting the evidence for these processes as involved in processing traumatic experience.”

As I noted above, this hypothesis states that a symptom of PTSD, IR, is a symptom of PTSD. I do not see the value of this hypothesis and finding.

RESPONSE: Amended - removed

30. Page 21

You wrote: “No association was found between PTG and gender, contradicting the existing literature (e.g. [21]) that suggests females are more likely to experience PTG.”

Again, since this finding is different than previous findings, it would be helpful for you to briefly explain why this is so.

RESPONSE: Amended lines 520-523

Such gender differences are more apparent with age, in that these differences increase as participants are older [23]. Given our sample were mostly young adults within a small age range, our distribution may be too narrow for such an effect. 

31. Page 22

Typo, line 472. Should be capitalized: “However, “

RESPONSE: Amended line 527

32. Under Limitations of this study, I think it is very important that you specify that you are studying Type 1 (simple trauma) and not Type 2 (complex trauma). I believe type 2 trauma is what most trauma therapists see and treat, more often.

RESPONSE: Amended line 532-533, 539

33. You wrote: “As Taylor[54] argues, psychology needs to find a scientific method of measuring value systems or it risks losing the human element from which it stemmed, especially in regards to the subject of trauma, when values and beliefs of this kind are often relied on for comfort and strength.”

This is a very important statement. Well done.

RESPONSE: Thank you. No amendment made. 

34. I think several of your clinical implications are of value. However, you wrote:

“Belief systems can be used as a framework and context in which to promote DR in order to derive some benefit from distress [55].”

You would need to flesh this out much more if you are to make this statement. Please explain how “belief systems” “promote DR”. At least offer one way this may be so, and indicate this can be further researched.

RESPONSE: Amended line 591-594. 

For example, a starting point from which to make sense of the world, find meaning in experience, find the positives in distressing experiences and grow and develop psychologically in the aftermath of trauma.

---

## [Decision Letter · Decision Letter 2]

28 Apr 2022

PONE-D-21-05935R2The role of repetitive thinking and spirituality in the development of posttraumatic growth and symptoms of posttraumatic stress disorderPLOS ONE

Dear Dr. Eames,

Thank you for submitting your manuscript to PLOS ONE. After careful consideration, we feel that it has merit but does not fully meet PLOS ONE’s publication criteria as it currently stands. Therefore, we invite you to submit a revised version of the manuscript that addresses the points raised during the review process. Please see some minor revisions suggested by the reviewer below, that you may incorporate in order to enhance clarity.

We look forward to receiving your revised manuscript.

Kind regards,

Hanna Landenmark

Staff Editor

PLOS ONE

Journal Requirements:

Reviewers' comments:

Reviewer's Responses to Questions

**Comments to the Author**

1. If the authors have adequately addressed your comments raised in a previous round of review and you feel that this manuscript is now acceptable for publication, you may indicate that here to bypass the “Comments to the Author” section, enter your conflict of interest statement in the “Confidential to Editor” section, and submit your "Accept" recommendation.

Reviewer #1: All comments have been addressed

2. Is the manuscript technically sound, and do the data support the conclusions?

Reviewer #1: Yes

3. Has the statistical analysis been performed appropriately and rigorously? 

Reviewer #1: I Don't Know

4. Have the authors made all data underlying the findings in their manuscript fully available?

Reviewer #1: Yes

5. Is the manuscript presented in an intelligible fashion and written in standard English?

Reviewer #1: Yes

6. Review Comments to the Author

Reviewer #1: Review of 2nd Revision (Repetitive Thinking) (KB, 2-12-22)

PLOS ONE

The role of repetitive thinking and spirituality in the development of posttraumatic growth and symptoms of posttraumatic stress disorder

PONE-D-21-05935R2

Page 10, lines 208-210

You wrote:

These were included as both factors were deemed important to include as potential confounders when measuring cognitive response to trauma.

Change wording to:

These were included as both factors were deemed important as potential confounders when measuring cognitive response to trauma.

Page 22, lines 428-432

Add parenthesis at end of sentence, as in:

In considering the items that comprise the subscale, items are phrased (reverse scored) with more of a focus on the individual alone rather than a broader sense of spirituality (“It always seems that I am doing things wrong”, “I am not comfortable with myself”, “Much of what I do in life seems strained”, “My life is often troublesome”, “I often feel tense”, and “I am an unhappy person”.)

Page 27, lines 547-548

Change wording, to make clearer the connection with the previous sentence, as in:

For example, belief systems can serve as a starting point from which to make sense of the world, find meaning in experience, find the…

7. PLOS authors have the option to publish the peer review history of their article (what does this mean?). If published, this will include your full peer review and any attached files.

Reviewer #1: No

---

## [Author Response · Author response to Decision Letter 2]

31 May 2022

Reviewer comments 

1. Page 10, lines 208-210. 

You wrote:

These were included as both factors were deemed important to include as potential confounders when measuring cognitive response to trauma. 

Change wording to: 

These were included as both factors were deemed important as potential confounders when measuring cognitive response to trauma.

Response: Wording changed as advised (deleted ‘to include’), page 10, line 209.

2. Page 22, lines 428-432

Add parenthesis at end of sentence, as in:

In considering the items that comprise the subscale, items are phrased (reverse scored) with more of a focus on the individual alone rather than a broader sense of spirituality (“It always seems that I am doing things wrong”, “I am not comfortable with myself”, “Much of what I do in life seems strained”, “My life is often troublesome”, “I often feel tense”, and “I am an unhappy person”.)

Response: Parenthesis included, page 22, line 433. 

3. Page 27, lines 547-548

Change wording, to make clearer the connection with the previous sentence, as in:

For example, belief systems can serve as a starting point from which to make sense of the world, find meaning in experience, find the… 

Response: Wording changed as suggested (include ‘belief systems can serve as…’), page 27, line 548-9.

---

## [Decision Letter · Decision Letter 3]

13 Jul 2022

The role of repetitive thinking and spirituality in the development of posttraumatic growth and symptoms of posttraumatic stress disorder

PONE-D-21-05935R3

Dear Dr. Eames,

We’re pleased to inform you that your manuscript has been judged scientifically suitable for publication and will be formally accepted for publication once it meets all outstanding technical requirements.

Kind regards,

George Vousden

Staff Editor

PLOS ONE

Additional Editor Comments (optional):

Reviewers' comments:

Reviewer's Responses to Questions

**Comments to the Author**

1. If the authors have adequately addressed your comments raised in a previous round of review and you feel that this manuscript is now acceptable for publication, you may indicate that here to bypass the “Comments to the Author” section, enter your conflict of interest statement in the “Confidential to Editor” section, and submit your "Accept" recommendation.

Reviewer #1: All comments have been addressed

2. Is the manuscript technically sound, and do the data support the conclusions?

Reviewer #1: (No Response)

3. Has the statistical analysis been performed appropriately and rigorously? 

Reviewer #1: (No Response)

4. Have the authors made all data underlying the findings in their manuscript fully available?

Reviewer #1: (No Response)

5. Is the manuscript presented in an intelligible fashion and written in standard English?

Reviewer #1: (No Response)

6. Review Comments to the Author

Reviewer #1: (No Response)

7. PLOS authors have the option to publish the peer review history of their article (what does this mean?). If published, this will include your full peer review and any attached files.

Reviewer #1: No

---

## [Editor Report · Acceptance letter]

27 Jul 2022

PONE-D-21-05935R3 

The role of repetitive thinking and spirituality in the development of posttraumatic growth and symptoms of posttraumatic stress disorder 

Dear Dr. Eames:

I'm pleased to inform you that your manuscript has been deemed suitable for publication in PLOS ONE. Congratulations! Your manuscript is now with our production department. 

Kind regards, 

on behalf of

Dr. George Vousden 

Staff Editor

PLOS ONE